# An experimental proposal to study collapse of the wave function in travelling-wave parametric amplifiers

T.H.A. van der Reep[1], L. Rademaker[2,3], X.G.A. Le Large[1], R.H. Guis[1] and
T.H. Oosterkamp[1*]

**1** Leiden Institute of Physics, Leiden University, Niels Bohrweg 2, 2333 CA Leiden, The Netherlands
**2** Department of Theoretical Physics, University of Geneva, 1211 Geneva, Switzerland
**3** Perimeter Institute for Theoretical Physics, Waterloo, Ontario N2L 2Y5, Canada
* oosterkamp@physics.leidenuniv.nl

May 27, 2020

## Abstract

**The read-out of a microwave qubit state occurs using an amplification chain that enlarges the quantum state to a signal detectable with a classical measurement apparatus. However, at what point in this process did we really 'measure' the quantum state? In order to investigate whether the 'measurement' takes place in the amplification chain, we propose to construct a microwave interferometer that has a parametric amplifier added to each of its arms. Feeding the interferometer with single photons, the visibility depends on the gain of the amplifiers and whether a measurement collapse has taken place during the amplification process. We calculate the interference visibility as given by standard quantum mechanics as a function of gain, insertion loss and temperature and find a magnitude of $1/3$ in the limit of large gain without taking into account losses. This number reduces to $0.26$ in case the insertion loss of the amplifiers is $2.2\,\mathrm{dB}$ at a temperature of $50\,\mathrm{mK}$. We show that if the wave function collapses within the interferometer, we will measure a reduced visibility compared to the prediction from standard quantum mechanics once this collapse process sets in.**

## 1 Introduction

When a photon hits a single-photon detector, for example a photo-multiplier tube (PMT), a chain of events is set in motion that would lead to an audible click or signal that can be processed by a classical observer. In the case of a PMT the photon is absorbed in the PMT's photo-cathode and, in turn, a photo-electron is emitted. The electron is amplified in several stages resulting in a detectable current pulse at the anode of the device.

A similar situation occurs for microwave photons in quantum bit (qubit) experiments [1]. The read-out of the qubit state, which can be prepared in a single-photon state [2], occurs via read-out lines that run from the device to the measurement apparatus. Implemented in the read-out lines is an amplification chain to enlarge the tiny qubit signal to human proportions.

It follows that a measurement can be seen as a process: A quantum signal enters a measurement device (to which we here count the amplification chain in case of qubit experiments), it is amplified and finally the apparatus is read-out. In this article we are interested in the question: at what point in the process did we really 'measure' the quantum state? When did the system change from being purely quantum-mechanical to classical?
We envision to probe the level of quantum coherence during amplification, by building an interferometer around two microwave parametric amplifiers. By comparing the measured interference pattern to the expected interference for a fully quantum-mechanical state, we can infer at which gain level we start deviating from this expectation. In the remainder of this article we will therefore compare interference visibilities for a quantum system to a system that experienced a spontaneous measurement within the interferometer in the Born sense.

The amplifiers we propose to use are typically used in the first amplification stage of qubit read-out lines, since they provide a large gain, are nearly quantum limited and can be described using conventional quantum theory [3–13]. Our experiments are partially inspired by similar set-ups with optical photons using non-linear optical parametric amplification, by e.g. Zeilinger [14] and De Martini [15], or other

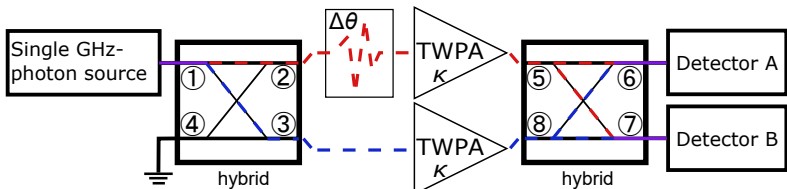

Figure 1: Schematic overview of a balanced microwave amplifier set-up. Using a 90°-hybrid (microwave analogue of a beam splitter), single photons are brought in a superposition, which is then amplified using two identical TWPAs, characterised by an amplification $\kappa$. Before entering the TWPAs, the excitation in the upper arm is phase shifted by $\Delta\theta$, which is assumed to account for all phase differences within the set-up. Using a second 90°-hybrid, we can study the output radiation from arms 6 and 7 using detectors A and B. Using 4WM TWPAs, an idler mode is generated. The interference of the idler mode can be studied indepently of the interference of the signal mode using the same detectors.

techniques by e.g. Gisin [16] and Rempe [17].

In this article we will not argue for one or the other possible mechanisms of the collapse process. The variety of possible ideas is large, see e.g. [18]. Instead, the work presented here only relies on Born's rule: the probability of a certain outcome after measurement is proportional to the wavefunction-squared.

In section 2 we calculate the Hamiltonian of the interferometer in the lossless case in the time domain. In section 3 we introduce a measure for the visibility of our interferometer and we discuss the theoretical predictions for this visibility as a function of the gain of the interferometers. In section 4 we discuss the effect of losses followed by our ideas on observing spontaneous collapse in section 5. In the final section we conclude by elaborating on the realisation of the experiment and estimating the feasibility of the experiment with parametric amplifiers with a gain of $40\,\mathrm{dB}$ – a gain commonly used to read out qubits in quantum computation experiments. Some of the detailed calculations are deferred to the supplementary information.

## 2    Model – lossless case

We consider the Mach-Zehnder type interferometer depicted in figure 1. The interferometer is fed by a single photon source (signal) in input 1 and a travelling-wave parametric amplifier (TWPA) is added to each of its arms. Although other realisations of the experiment are conceivable, we argue in the supplementary material why we view this version as optimal (see appendices A and B). The signal enters a hybrid (the microwave analogue to a beam splitter), thereby creating a superposition of 0 and 1 photons in each of the arms. The excitation in the upper arm of the interferometer can be phase shifted, where we assume that the phase shift accounts for an intended phase shift as well as all unwanted phase shifts due to fabrication imperfections and the non-linear phase shift from the TWPA. In the TWPA amplification takes place by a wave mixing interaction. Throughout the paper we use TWPAs working by a four-wave mixing (4WM) process in a mode which is phase preserving (i.e. the amplification is independent of the pump phase) and non-degenerate (i.e. the pump and the signal are at different frequencies $\omega_\mathrm{p}$ and $\omega_\mathrm{s}$, respectively). We assume the pump to be degenerate (one signal photon is created by destroying two pump photons and by energy conservation this gives rise to an idler at frequency $\omega_\mathrm{i} = 2\omega_\mathrm{p} - \omega_\mathrm{s}$). We also assume that the pump is undepleted (we neglect the decrease of pump photons in the amplification process). Finally, we assume that the pump, signal and idler are phase-matched ($2k_\mathrm{p} = k_\mathrm{s} + k_\mathrm{i}$, where $k$ is the wave number including self- and cross-modulation due to the non-linear wave mixing). After the TWPA, the excitations from the two arms are brought together using another hybrid and we can study the output radiation in both the signal and idler mode with detectors A and B.

In this section we ignore losses, the effect of which we will discuss in section 4. Under the assumptions assumptions introduced above [12, 13]

$$\hat{H}_{\mathrm{TWPA}} = -\hbar\chi \left( \hat{a}_\mathrm{s}^\dagger \hat{a}_\mathrm{i}^\dagger + \mathrm{H.c.} \right). \tag{1}$$

Here $\hbar$ is the reduced Planck constant $h/2\pi$ and $\chi$ is the non-linear coupling derived from the third-order susceptibility of the transmission line, which takes into account the pump intensity. $\hat{a}_n^\dagger$ is the creation

operator of mode $n$. Using the Heisenberg equations of motion, one can solve for the evolution of the annihilation operators analytically. This yields [12]

$$\hat{a}_{\text{s(i)}}(t) = \hat{a}_{\text{s(i)}}(0)\cosh\kappa + i\hat{a}^{\dagger}_{\text{i(s)}}(0)\sinh\kappa, \tag{2}$$

where $\kappa \equiv \chi\Delta t_{\text{TWPA}}$ is the amplification if the state spends a time $\Delta t_{\text{TWPA}}$ in the TWPA. Thus, we can determine the average number of photons in the signal (idler) mode as function of the amplification of the amplifier as

$$\langle \hat{n}_{\text{s(i)}} \rangle_{\text{out}} = \langle \hat{n}_{\text{s(i)}} \rangle_{\text{in}}\cosh^2\kappa + \left(\langle \hat{n}_{\text{i(s)}} \rangle_{\text{in}} + 1\right)\sinh^2\kappa \tag{3}$$

provided that the signal and/or idler are initially in a number state. $\langle \hat{n} \rangle_{\text{out (in)}}$ is the average number of photons leaving (entering) the TWPA. From this relation we define the amplifier gain as $G_{\text{s}} = \langle \hat{n}_{\text{s}} \rangle_{\text{out}} / \langle \hat{n}_{\text{s}} \rangle_{\text{in}}$.

Even though under these assumptions the calculation can be done analytically (see Appendix C) we present the numerical implementation here, because to such an implementation losses can be added straightforwardly at a later stage.

To numerically obtain the output state we use QuTiP [19]. We first split the Hilbert space of the interferometer into the upper arm and the lower arm. Each of the arm subspaces is additionally divided into a signal and idler subspace. Hence, our numerical Hilbert space has dimension $N^4$, where $N-1$ is the maximum amount of signal and idler photons taken into account in each of the arms. In this framework the input state is

$$|\psi\rangle = |1\rangle_{\text{up,s}}|0\rangle_{\text{up,i}}|0\rangle_{\text{low,s}}|0\rangle_{\text{low,i}}, \tag{4}$$

where the labels 'up' and 'low' refer to the upper and lower arm of the interferometer respectively. We evolve this state by the time evolution operator, generated by the Hamiltonian $\hat{H}$ of the system. The first hybrid is described by the Hamiltonian

$$\hat{H}_{\text{h1}} = -\frac{\hbar\pi}{4\Delta t_{\text{h1}}}\left(\sum_{n=\text{s,i}}\hat{a}^{\dagger}_{\text{up},n}\hat{a}_{\text{low},n} + \text{H.c.}\right). \tag{5}$$

where $\Delta t_{\text{h1}}$ is the time spent in the hybrid. Note that state evolution with the above Hamiltonian for a time $\Delta t_{\text{h1}}$ corresponds to the transformation operator for an ordinary 90°-hybrid,

$$\hat{U}_{\text{h1}} = e^{i\hat{H}_{\text{h1}}\Delta t_{\text{h1}}/\hbar} = e^{i\frac{\pi}{4}\left(\sum_{n=\text{s,i}}\hat{a}^{\dagger}_{\text{up},n}\hat{a}_{\text{low},n}+\text{H.c.}\right)}. \tag{6}$$

By the same reasoning, the Hamiltonian of the phase shifter can be written as

$$\hat{H}_{\text{ps}} = \frac{\hbar\Delta\theta}{\Delta t_{\text{ps}}}\left(\sum_{n=\text{s,i}}\hat{a}^{\dagger}_{\text{up},n}\hat{a}_{\text{up},n} + \text{H.c.}\right), \tag{7}$$

where $\Delta\theta$ is the applied phase shift. In our numerical calculations we use

$$\hat{H}^{(\text{up/low})}_{\text{TWPA}} = -\frac{\hbar\kappa_{(\text{up/low})}}{\Delta t_{\text{TWPA}}}\left(\hat{a}^{\dagger}_{(\text{up/low}),\text{s}}\hat{a}^{\dagger}_{(\text{up/low}),\text{i}} + \text{H.c.}\right) \tag{8}$$

for the TWPAs. After the TWPAs, the excitations from the two arms are brought together using a second hybrid to create interference, which is measured with detectors A and B. The second hybrid is described by a Hamiltonian $\hat{H}_{\text{h2}}$ similar to equation (5).

To summarize, the proposed theoretical model of the experiment in the absence of losses is as follows. We start with an initial single signal photon in the upper arm, described by equation (4). We evolve this state for a time $\Delta t_{h1}$ with Hamiltonian $\hat{H}_{\text{h1}}$, followed by $\hat{H}_{\text{ps}}$ for a time $\Delta t_{\text{ps}}$, then for a time $\Delta t_{\text{TWPA}}$ with $\hat{H}_{\text{TWPA}}$ of equation (8) and finally for a time $\Delta t_{h2}$ with Hamiltonian $\hat{H}_{\text{h2}}$. Finally, we will measure the photon densities in detector A and B, which leads to a given visibility of the interference pattern. For the loss-less case the values of the various $\Delta t$s can be chosen arbitrarily.

# 3    Interference visibility

From the state resulting from our calculations we get the probability distribution of number states in the detectors A and B, $Pr(n_{A,s}=i, n_{A,i}=j, n_{B,s}=k, n_{B,i}=l)$, from which we can calculate the photon number statistics and correlations by performing a partial trace (see appendix D). From the photon number statistics we can compute the visibility of the interference pattern. Although microwave photon counters have been developed in an experimental setting [20–22], we can also envision the measurement of the output radiation using spectrum analysers. Such instruments measure the output power, $P$, of the interferometer as a function of time and one can determine the number of photons arriving in the detectors as

$$n = \frac{1}{\hbar\omega} \int_{t_1}^{t_2} P(t)\, \mathrm{d}t. \tag{9}$$

Measuring the average photon number at detectors A and B, we can define the interference visibility as (appendix E)

$$V_{s(i)} \equiv \left. \frac{\langle n_{B,s(A,i)} \rangle - \langle n_{A,s(B,i)} \rangle}{\langle n_{B,s(A,i)} \rangle + \langle n_{A,s(B,i)} \rangle} \right|_{\Delta\theta=0}. \tag{10}$$

In case the amplifiers have an identical gain, the calculation can be simplified. Then, the visibility can be calculated using a smaller Hilbert space by the following observation: a single TWPA fed with a $|1\rangle_s |0\rangle_i$-state yields the average number of signal (idler) photons in detector B (A) as calculated with equation (3). Contrarily, feeding this TWPA with a $|0\rangle_s |0\rangle_i$-state gives the average number of signal (idler) photons in detector A (B) (see appendix F). This provides a reduced Hilbert space that scales as $2N^2$ for calculating the average visibility. Moreover, this observation implies that the visibility can be computed directly by substitution of equation (3) into equation (10).
Therefore, the visibility in the lossless case can be solved exactly. Regardless of the input, the parametric amplifier always outputs $\sinh^2 \kappa$ extra photons. In the case of an initial single-photon state, the extra term $\cosh^2 \kappa$ should be added. Consequently, the signal visibility becomes

$$V_s = \frac{\cosh^2 \kappa}{\cosh^2 \kappa + 2\sinh^2 \kappa}. \tag{11}$$

In the limit of large gain, the sinh and cosh become equal in magnitude, and consequently the visibility tends to 1/3. Similarly, the idler photon number will be $2\sinh^2 \kappa$ in the arm with an initial signal photon and $\sinh^2 \kappa$ in the other, consequently the idler visibility is constant at 1/3. The reduction from 1 to 1/3 is thus completely due to the addition of extra photons by the paramp.

The result of the calculations of the signal and idler visibilities are shown in figure 2 (in red) and have been verified using our analytical results from Appendix C up to $\kappa = 0.8$ and our numerical results up to $\kappa = 1.7$. It shows that the signal interference visibility drops from 1 to 1/3 with increasing gain, in accordance with [23]. The signal visibility at $\kappa = 0$ is 1, since this situation resembles an ordinary single photon interferometer. The idler visibility at $\kappa = 0$ is undefined due to the absence of idler photons. Please note that a superposition of zero and one photon before an amplifier with gain G, does not result in a superposition of zero and G photons after the amplifier. To emphasize that this results in multi-photon interference we present a figure in appendix D that shows the photon number correlations within the interferometer arms. Furthermore, this figure shows how many photon Fock states are involved for different gain of the amplifiers.

# 4    The effect of losses

To take into account the effect of losses (dissipation/insertion loss) we use the Lindblad formalism, which provides the expression for the time evolution of the density matrix, $\hat{\rho}$ [24],

$$\frac{\mathrm{d}\hat{\rho}}{\mathrm{d}t} = -\frac{i}{\hbar}[\hat{H}, \hat{\rho}] + \sum_{n=1}^{N^2-1} \left( \hat{J}_n \hat{\rho} \hat{J}_n^\dagger - \frac{1}{2}\left\{ \hat{\rho}, \hat{J}_n^\dagger \hat{J}_n \right\} \right) \tag{12}$$

where $\{\,,\}$ denotes the anticommutator and $\hat{J}_n$ are the jump operators. These operators describe transitions that the system may undergo due to interactions with the surrounding thermal bath. Losses can be

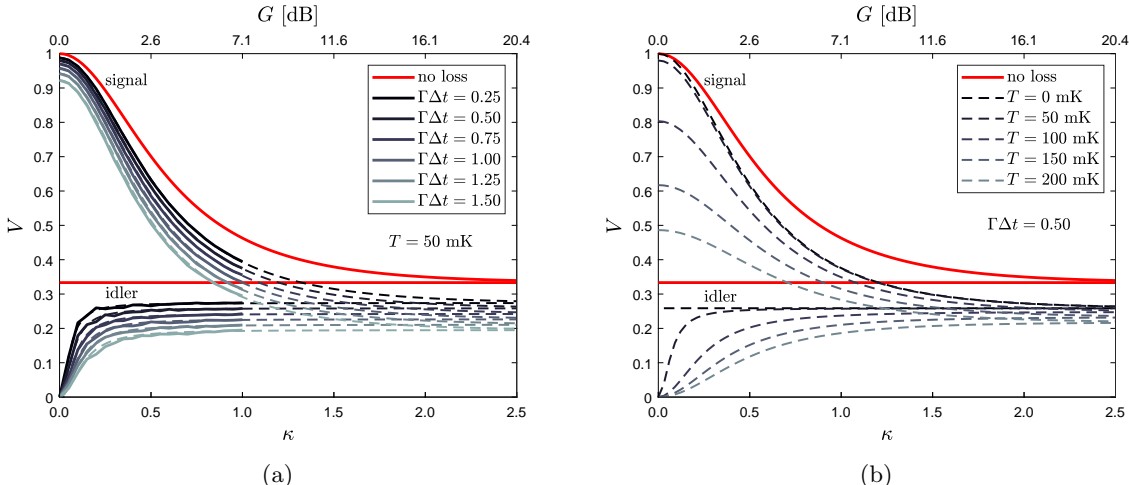

Figure 2: Expected visibility of the interference pattern of the interferometer as a function of amplification $\kappa$ for signal and idler using the reduced Hilbert space (see text). The gain in dB on the upper axis is only indicative and does not take into account the losses in the amplifiers ($G = 10 \log_{10} \langle n_s \rangle_{\text{out}} / \langle n_s \rangle_{\text{in}} = 10 \log_{10} \cosh \kappa + 2 \sinh \kappa$). Without loss (red) the visibility tends to $1/3$ for large gain. The visibility in case losses are added to the system is plotted in grey for various amounts of loss in the TWPAs at (a) $T = 50\,\text{mK}$ ($n_{\text{th}} = 8.3 \times 10^{-3}$) varying $\Gamma \Delta t_{\text{TWPA}}$ ($\Gamma = 100\,\text{MHz}$, loss $\approx 4 \Gamma \Delta t\,[\text{dB}]$) and (b) $\Gamma \Delta t_{\text{TWPA}} = 0.50$ ($\Gamma = 100\,\text{MHz}$) varying $T$. For each of the hybrids and the phase shifter the loss is set to $\Gamma \Delta t = 0.1$ and we have set $\omega_{\text{s,i}} = 2\pi \times 5\,\text{GHz}$. The reduced Hilbert space calculations are presented in continous lines, whereas an analytical fit and extrapolation according to equation (16) is dashed. We find that even TWPA losses as high as $6\,\text{dB}$ do not reduce the visibility to 0.

described by the jump operators $\hat{J}_{\text{out}}$ and $\hat{J}_{\text{in}}$. $\hat{J}_{\text{out}}$ describes a photon leaving the system and entering the bath,

$$\hat{J}_{\text{out},n} = \sqrt{\Gamma \left(1 + n_{\text{th}}\right)} \hat{a}_n, \tag{13}$$

where $\Gamma$ is the loss rate and $n_{\text{th}} = 1/(\exp(\hbar \omega / k_{\text{B}} T) - 1)$ is the thermal occupation number of photons in the bath. $\hat{J}_{\text{in}}$ describes a photon entering the system from the bath,

$$\hat{J}_{\text{in},n} = \sqrt{\Gamma n_{\text{th}}} \hat{a}_n^\dagger. \tag{14}$$

Here we again see the advantage of using a description in the time domain and putting $\Delta t$ in the component Hamiltonians, equations (5), (7) and (8), in section 2 the total (specified) loss is mainly determined by the product $\Gamma \Delta t$ relating to the (insertion) loss as

$$\begin{aligned} IL &= -10 \log_{10} \left( (1 - n_{\text{th}} / \langle n_{\text{in}} \rangle) \, e^{-\Gamma \Delta t} + n_{\text{th}} / \langle n_{\text{in}} \rangle \right) \\ &\approx 4 \Gamma \Delta t. \end{aligned} \tag{15}$$

The approximation holds for $n_{\text{th}}$ small. This allows us to define a constant loss rate for the whole set-up, while adjusting $\Delta t$ for each component to match the actual loss. Since the photon state in the interferometer is now described by a density matrix, the amount of memory for these calculations scales as $N^8$.

To study the effect, we set $\omega_{\text{s,i}} = 2\pi \times 5\,\text{GHz}$ for now. The loss rate $\Gamma$ is set to $100\,\text{MHz}$ for the full set-up. For the hybrids and the phase shifter, we choose $\Delta t_{(\text{h}_1,\text{ps},\text{h}_2)} = 1\,\text{ns}$ ($IL \approx 0.4\,\text{dB}$) and study the effect of losses in the TWPAs by varying $\Delta t_{\text{TWPA}}$ and $T$. We evolve the state under the Hamiltonians $\hat{H}_{\text{h}_1} \rightarrow \hat{H}_{\text{ps}} \rightarrow \hat{H}_{\text{TWPA}}^{(\text{up/low})} \rightarrow \hat{H}_{\text{h}_2}$ as described in section 2.
Unfortunately, running the numeric calculation, we were not able to increase the amplification to $\kappa > 0.6$ due to QuTiP working with a version of SciPy supporting only int32 for element indexing. However, again it appears that we can use the method of the reduced Hilbert space sketched in the last section. Thus, the problem only scales as $2N^4$, and we have performed the numeric calculation up to $\kappa = 1.0$. Applying the reduced Hilbert space approach, we found that the parametric amplifier's output in presence of losses can be fitted according to

$$\langle \hat{n}_{\text{s(i)}} \rangle_{\text{out}} = \langle \hat{n}_{\text{s(i)}} \rangle_{\text{out}}|_{\kappa=0} \cosh^2 \kappa + \left( \langle \hat{n}_{\text{i(s)}} \rangle_{\text{out}}|_{\kappa=0} + 1 \right) e^{-f} \sinh^2 \kappa \tag{16}$$

where the parameter $f$ depends on $\Gamma$, the various $\Delta t$s (if $T > 0$), $n_{\mathrm{th}}$ and the input state and is determined by a fit to the numerical data (see Appendix G). $\langle \hat{n}_{\mathrm{s(i)}} \rangle_{\mathrm{out}|_{\kappa=0}}$ is the number of photons leaving the amplifier in case no amplification is present,

$$\langle \hat{n}_{\mathrm{s(i)}} \rangle_{\mathrm{out}} |_{\kappa=0} = \left( \langle \hat{n}_{\mathrm{s(i)}} \rangle_{\mathrm{in}} - n_{\mathrm{th}} \right) \mathrm{e}^{-\Gamma \Delta t_{\mathrm{tot}}} + n_{\mathrm{th}}. \tag{17}$$

This allows us to extrapolate the results to higher gain.

The results of the calculations with loss are also depicted in figure 2 assuming the full set-up is at a constant temperature. We observe that losses decrease the interference visibility with respect to the case where losses were neglected. However, even for TWPA losses as high as $6\,\mathrm{dB}$, the interference visibility survives. As in the no-loss case the signal and idler visibility converge asymptotically to the same value. In the high-gain limit, the interference visibility is given by

$$V_{\mathrm{s,i}} = \left( 1 + 2\mathrm{e}^{\Gamma \Delta t_{\mathrm{tot}} - f} + 2n_{\mathrm{th}}\mathrm{e}^{\Gamma \Delta t_{\mathrm{tot}}} \left( 1 + \mathrm{e}^{-f} \right) \left( 1 - \mathrm{e}^{-\Gamma \Delta t_{\mathrm{tot}}} \right) \right)^{-1} \tag{18}$$

by equation (16). Assuming $n_{th} \ll 1$, we find $f \approx \Gamma \Delta t_{\mathrm{tot}}/2$ (see appendix G) and as a result

$$V_{\mathrm{s,i}} \approx \frac{1}{1 + 2\mathrm{e}^{\Gamma \Delta t_{\mathrm{tot}}/2}}. \tag{19}$$

Thus, in the limit of low temperature we find that the interference disappears exponentially with the loss in the set-up. The visibility becomes $1/e$ times the lossless visibility at $\Gamma \Delta t_{\mathrm{tot}} = 3$ ($IL \approx 12\,\mathrm{dB}$, but at this loss it will not be possible to keep the amplifiers in the limit of low $n_{\mathrm{th}}$).

Contrarily, in the limit of low losses, we find that $f \approx 0$ and

$$V_{\mathrm{s,i}} \approx \frac{1}{3 + 4n_{\mathrm{th}}\Gamma \Delta t_{\mathrm{tot}}}. \tag{20}$$

Thus, we see that the interference visibility becomes $1/e$ times the lossless visibility when approximately 1 photon jumps from the bath into the system.

Experimentally, the conclusion is that efforts need to be made to make the losses in the parametric amplifier so small, that the amplifier remains cold.

# 5    Observing collapse

Although there is currently no universally agreed-upon model that describes state collapse, we propose to mathematically investigate the effect of collapse on the proposed experiment using Born's rule in the following way.

To model the collapse we split each of the amplifiers in the upper and lower arm of the interferometer in two parts and we assume that the collapse takes place instantaneously in between these two parts, see figure 3. Thus, the first part of each amplifier can be characterised by a coupling constant $\eta\kappa$ and the second by a coupling constant $(1 - \eta)\kappa$, where $\eta \in [0, 1]$ sets the collapse position. If $\eta = 0$ the collapse takes place between the first hybrid and the amplifiers, while for $\eta = 1$ the collapse takes place between the amplifiers and the second hybrid. For $0 < \eta < 1$ the collapse takes place within the amplifiers. For simplicity, we ignore the fact that a photon is a spatially extended object. Moreover, we will ignore here that the collapse process might be expected to be stochastic in its position $\eta$, a point we will return to in section 6.

Then, by Born's rule we have to assume a collapse phenomenology. Regardless of the precise mechanism, such a collapse will destroy the entanglement between the two interferometer arms and yield a classical state. As for the type of classical state, we will consider two options: the state collapses onto (1) a number state, or (2) onto a coherent state. For both these options we will study the effect on the interference visibility below.

## 5.1    Collapse onto a number state

In case the collapse projects the instantaneous state onto a number state, the state after projection is given by $|\psi_{\mathrm{coll}}\rangle (N, M) = |N + 1\rangle_{\mathrm{up,s}} |N\rangle_{\mathrm{up,i}} |M\rangle_{\mathrm{low,s}} |M\rangle_{\mathrm{low,i}}$ or $|\psi_{\mathrm{coll}}\rangle (N, M) = |N\rangle_{\mathrm{up,s}} |N\rangle_{\mathrm{up,i}} |M + 1\rangle_{\mathrm{low,s}} |M\rangle_{\mathrm{low,i}}$, depending on whether the initial photon went through the upper or lower arm of the interferometer. Hence, this collapse phenomenology can be thought of as resulting from the collapse taking place as a

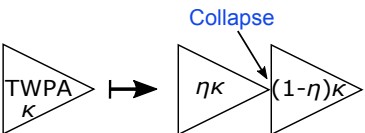

Figure 3: Model of a TWPA in which a quantum state collapse takes place. The quantum TWPA, characterised by coupling constant $\kappa$ is split in two parts. One is characterised by the coupling constant $\eta\kappa$ and the other by $(1-\eta)\kappa$, where $\eta \in [0,1]$ determines the position of the collapse. We assume that the state collapse takes place instantaneously between the two parts of the amplifier.

consequence of a which-path detection within the amplifiers, which would happen in a power meter that measures the intensity (energy) of an incoming signal. The second part of the amplifiers, characterised by the coupling constant $(1-\eta)\kappa$, evolves $|\psi_{\text{coll}}\rangle$ to $|\psi'_{\text{coll}}\rangle = \sum_{N,M} c_{NM} |\psi_{\text{coll}}\rangle (N,M)$, where $c_{NM}$ are the weights determined by $(1-\eta)\kappa$ and $\sum_{N,M} |c_{NM}|^2 = 1$. $|\psi'_{\text{coll}}\rangle$ is the state just before the second hybrid.

To determine the effect on the interference visibility of such a collapse, we calculate $\langle n \rangle_{X,n} = \hat{a}^\dagger_{X,n} \hat{a}_{X,n}$, the number of photons arriving in detector $X \in \{A,B\}$ in mode $n \in \{s,i\}$. This equation can be rewritten in terms of creation and annihilation operators of the upper and lower arm of the interferometer by the standard hybrid transformation relations $\hat{a}_{[A]\{B\},n} \mapsto (\{1\}[i]\hat{a}_{\text{up},n} + \{i\}[1]\hat{a}_{\text{low},n})/\sqrt{2}$ to find

$$V_n^{\text{coll}} = \frac{i \langle \hat{a}^\dagger_{\text{up},n} \hat{a}_{\text{low},n} - \hat{a}_{\text{up},n} \hat{a}^\dagger_{\text{low},n} \rangle}{\langle \hat{a}^\dagger_{\text{up},n} \hat{a}_{\text{up},n} + \hat{a}^\dagger_{\text{low},n} \hat{a}_{\text{low},n} \rangle}, \tag{21}$$

which equals 0 for any $|\psi'_{\text{coll}}\rangle$. Hence, we find that a collapse onto a number state within the interferometer causes a total loss of interference visibility.

## 5.2 Collapse onto a coherent state

If a collapse in the amplifiers projects the quantum state onto a coherent state, the state after collapse is $|\psi_{\text{coll}}\rangle = |\alpha_{\text{up,s}}\rangle |\alpha_{\text{up,i}}\rangle |\alpha_{\text{low,s}}\rangle |\alpha_{\text{low,i}}\rangle$ with overlap $c_{\text{coll}} = \langle \psi_{\text{coll}}|\psi_{\text{q}}\rangle$. Here $|\psi_{\text{q}}\rangle$ is the instantaneous quantum state at the moment of collapse. This collapse phenomenology can be thought of as the electrons in the transmission lines connecting the different parts of the interferometer collapsing into position states characterised by a well-defined phase and amplitude contrary to their ill-defined phase and amplitude in case the transmission lines are excited with a (superposition of) photonic number states. Such a collapse happens in a vector network analyser, which measures both the intensity as well as the phase of an incoming signal.

In this case, the second part of the parametric amplifiers characterised by $(1-\eta)\kappa$ evolves the amplitudes $\alpha$ in $|\psi_{\text{coll}}\rangle$ into average amplitudes

$$\bar{\alpha}_{\text{up(low)},\text{s(i)}} = \alpha_{\text{up(low)},\text{s(i)}} \cosh(1-\eta)\kappa + i\alpha^*_{\text{up(low)},\text{i(s)}} \sinh(1-\eta)\kappa \tag{22}$$

by equation (2). Then the number of photons arriving in each detector is, for each individual collapse,

$$n_{\text{A(B)},n}^{\text{coll}} = \frac{1}{2} \left( |\bar{\alpha}_{\text{up},n}|^2 + |\bar{\alpha}_{\text{low},n}|^2 \mp 2|\bar{\alpha}_{\text{up},n}||\bar{\alpha}_{\text{low},n}| \sin(\phi_{\text{low},n} - \phi_{\text{up},n}) \right) \tag{23}$$

where $\phi_i$ is the phase of the state $\bar{\alpha}_i$. Thus, we can obtain the average number of photons arriving in each detector as an integration over all possible collapsed states weighed by their probability. That is

$$\langle n_{X,n}^{\text{coll}} \rangle = \frac{1}{\pi^4} \int n_{X,n}^{\text{coll}} |c_{\text{coll}}|^2 \text{d}^2\alpha_{\text{up,s}} \text{d}^2\alpha_{\text{up,i}} \text{d}^2\alpha_{\text{low,s}} \text{d}^2\alpha_{\text{low,i}} \tag{24}$$

in which $\text{d}^2\alpha_n$ denotes the integration over the complex amplitude of the coherent state $n$. Then, we determine the interference visibility according to equation (10).

In case we assume that the interferometer is lossless, we can perform such a calculation analytically (see appendix H). The resulting interference visibility is plotted in figure 4 in which we can observe that the interference visibility at high gain depends on the location of collapse. For $\eta = 1$ the signal and idler visibility equals $1/3$. For $\eta = 0.5$ both visibilities tend to approximately $0.15$ and in case $\eta = 0$ the visibility tends to $1/5$ for both signal and idler at high gain.

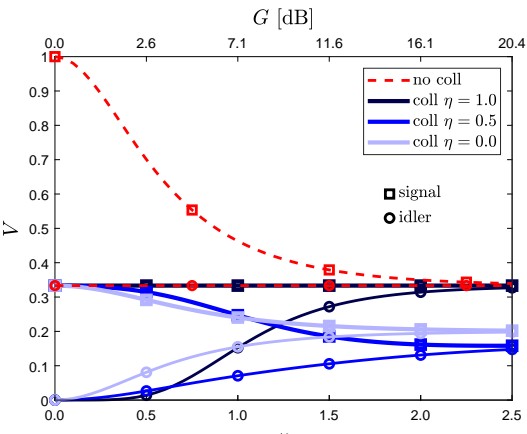

Figure 4: Comparison of the interference visibility resulting from a full quantum calculation without collapse and under the assumption of state collapse to coherent states within the interferometer assuming no losses. If the state collapses between the amplifiers and the second hybrid ($\eta = 1$), the visibility is 1/3 for the signal and rises to 1/3 with increasing amplification for the idler. In case the collapse takes place halfway through the amplifiers ($\eta = 0.5$), the visibility tends to 0.15 for both signal and idler for high gain and if the collapse is between the first hybrid and the amplifiers ($\eta = 0$), the visibility goes to 0.2 for signal and idler.

# 6 Experimental realisation and feasibility

As a single photon source, we propose to use a qubit capacitively coupled to a microwave resonator [2]. For the amplifiers we can use TWPAs in which the non-linearity is provided by Josephson junctions. Currently, TWPAs providing 20 dB ($\kappa = 2.5$) of gain and 2 dB of (insertion) loss that operate at $T = 30$ mK have been developed [9].

The amplification process within the TWPAs is driven by a coherent pump signal. Instead of increasing the gain of the TWPAs by increasing the pump power, we propose to vary the amplification by varying the pump frequency. In the latter method the amplification varies due to phase matching conditions within the amplifier. The advantage is that in this manner the transmission and reflection coefficients of the TWPA, which depend on the pump power [25], can be kept constant while varying the gain in the interferometer. Although we assumed perfect phase matching in the amplifiers for the results shown in this paper, we do not expect a large difference if one changes from a varying pump-power approach to a varying phase-matching approach.

Our calculations are based on a Taylor expansion up to the third-order susceptibility of a parametric amplifier. Typically, microwave TWPAs work close to the critical current of the device, such that this assumption might break down and we need to take into account higher orders as well. For TWPAs based on Josephson junctions, we can estimate as follows at which current a higher order Taylor expansion would become necessary.
In the Hamiltonian of a TWPA with Josephson junctions the non-linearity providing wave mixing arises from the Josephson energy

$$E_{\mathrm{J}} = I_{\mathrm{c}}\varphi_0 \left( 1 - \cos\left(\frac{\Phi}{\varphi_0}\right) \right) = I_{\mathrm{c}}\varphi_0 \sum_{n=1}^{\infty} \frac{(-1)^{n-1}}{(2n)!} \left(\frac{\Phi}{\varphi_0}\right)^{2n}. \tag{25}$$

Here, $I_{\mathrm{c}}$ is the junction's critical current and $\varphi_0$ is the reduced flux quantum $\Phi_0/2\pi$. Hence, the second-order ($n = 3$) non-linear effects have a factor $4!(\Phi_{\mathrm{p}}/\varphi_0)^2/6!$ smaller contribution than the first-order non-linear effects. This contribution causes the generation of secondary idlers and additional modulation effects. If we require that this contribution is less than 5% of the energy contribution of the first-order non-linear terms, we can estimate that the theory breaks down at $\Phi_{\mathrm{p}}/\varphi_0 \approx 1.2$ ($I_{\mathrm{p}}/I_{\mathrm{c}} \approx 0.78$). It is only in the third-order non-linearity that terms proportional to $(\hat{a}_{\mathrm{s}}^\dagger \hat{a}_{\mathrm{i}}^\dagger)^n$ with $n > 1$ start to appear, apart from yet additional secondary idlers and further modulation effects. These terms have a maximal contribution of approximately a factor $4!(\Phi_{\mathrm{p}}/\varphi_0)^4/8! \approx 4 \times 10^{-3}$ less than the first-order non-linear term at the critical flux ($\Phi_{\mathrm{p}}/\varphi_0 = \pi/2$) and are therefore negligible for practical purposes.

The other assumption that might break down is the assumption of an undepleted pump. If the signal power becomes too close to the pump power, the pump becomes depleted. Typically this happens at $P_s \approx P_p/100$ [25]. At $I_p/I_c = 0.9$, $P_p \approx 1\,\text{nW}$ in a $50\,\Omega$-transmission line with $I_c = 5\,\mu\text{A}$. In case our qubit photon source has a $T_1$ time of approximately $100\,\text{ns}$ [2], implying the photon has a duration in that order, the number of $5\,\text{GHz}$-pump photons available for amplification is in the order of $10^7$. Hence, we expect that pump depletion only starts to play a significant role in case the amplification becomes about $50\,\text{dB}$.

In our calculations the only loss-effect that was not taken into account was the loss of pump photons due to the insertion loss of the TWPA. If the insertion loss amounts to $3\,\text{dB}$, half of the pump photons entering the device will be dissipated. To our knowledge, this effect has not been considered in literature. However, effectively this must lead to a non-linear coupling constant $\chi$ (equation (1)), which decreases in magnitude in time. In a more involved calculation this effect needs to be taken into account for a better prediction of the experimental outcome of the visibility.

Apart from making $\chi$ time dependent, the loss of pump photons will be the main reason for an increase of temperature of the amplifiers. A dilution refrigerator is typically able to reach temperatures of $10\,\text{mK}$ with a cooling power of $1\,\mu\text{W}$. However, the heat conductivity of the transmission line to the cold plate of the refrigerator will limit the temperature of the TWPA. Still, we estimate that a dissipation in the order of $0.5\,\text{nW}$ will not heat up the amplifiers above $50\,\text{mK}$. However, as shown in figure 2, even if the amplifiers heat up to temperatures as large as $200\,\text{mK}$ we still expect a visibility that should be easily measurable, if no collapse would occur.

Finally, a more accurate calculation of the expected interference visibility would need to take into account reflections within the set-up as well as the possible difference in gain between both amplifiers and decoherence mechanisms that might be present and we have not considered here, such as pure dephasing.

The results we obtained for the interference visibility with a collapse within the interferometer are only speculative as the mechanism of state collapse is currently not understood. In case the state collapses onto a number state, the resulting interference visibility is 0 for any gain. We anticipate that this number might increase in case losses are taken into account in the calculation, however, still we expect that the difference in interference visibility between the cases of no collapse and collapse within the interferometer should be easily detectable.

Contrarily, if the state collapses onto a coherent state, the visibility depends on the location of the collapse. This result should be interpreted as follows. Let us assume that the state collapses at a gain of $20\,\text{dB}$ ($\kappa = 2.5$). Then, neglecting losses, the predicted signal interference visibility is approximately $1/3$ in case the state does not collapse, whereas it equals $1/3$ in the case the state collapses between the amplifiers and the second hybrid ($\eta = 1$). However, if we increase the gain further, the expected location of collapse (the location at which the state is amplified by $20\,\text{dB}$) moves towards the first hybrid ($\eta < 1$), which will become apparent in the measurement result as an initial gradual drop in the interference visibility followed by an increase, see figure 4. Simultaneously, the idler visibility is expected to show the same behaviour.

It should be noted that the result for a calculation, in which one assumes a state collapse onto a coherent state between the interferometer and the detectors, is the same as when the state would collapse between the amplifiers and the second hybrid of the interferometer. However, even if this would be the case, one can observe a collapse within the interferometer if the collapse takes place within the amplifiers. A second remark to this collapse phenomenology is that it does not conserve energy. If one considers some state $|\psi\rangle$ with an average photon number $n$, one finds that a collapse onto a coherent state adds one noise photon to the state, i.e. $\langle n \rangle \mapsto \langle n \rangle^{\text{coll}} = n + 1$. This behaviour holds for each of the Hilbert subspaces. Such an increase in energy is a property of many spontaneous collapse models [26–30].

It is due to this added photon and its amplification (see equation (22)) in the classical part of the TWPAs that the differences in the predicted interference visibility with and without state collapse arise, although in the collapse the phase correlations between the signal and idler modes in both arms are preserved. The latter can be observed in our expression for $c_{\text{coll}}$ in appendix H. In case the photon is added after the amplifiers ($\eta = 1$) this photon can be added directly to the expression for the number of output photons (equation (3)), such that the expression for the interference visibility (equation (10)) goes from $V_s = \cosh^2 \kappa / (\cosh^2 \kappa + 2\sinh^2 \kappa)$ to $V_s^{\text{coll}} = \cosh^2 \kappa / (\cosh^2 \kappa + 2\sinh^2 \kappa + 2) = 1/3$ using the reduced Hilbert space approach. In case the state collapses before the amplifiers ($\eta = 0$) this photon can be added to $\langle \hat{n}_{s(i)} \rangle$ in equation (3) directly. Then, since the amplifiers are in this case fully classical, one can drop the $+1$ in the term $(\langle \hat{n}_{i(s)} \rangle + 1)$ in this equation, which results from the commutator $[\hat{a}, \hat{a}^\dagger] = 1$

. Such it is found that the interference visibility reduces to $V_\mathrm{s}^\mathrm{coll} = \cosh^2 \kappa/(3\cosh^2 \kappa + 2\sinh^2 \kappa)$, which equals $1/5$ in the high-gain limit.

In case one assumes a collapse onto a coherent state one could calculate the expected interference visibility in case losses are included numerically by calculating the overlap between the state evolved until collapse and many (order $10^6$) randomly chosen coherent states. However, due to the issue with SCIPY noted in section 4, we could not perform this calculation for a reasonable number of photons. Still we expect that, although the difference in visibility between the situations with and without collapse in the interferometer might be decreased, this difference is measurable.

Finally, as remarked in section 5, it might be expected that the collapse will take place at a position $\eta$, which is stochastic in nature. In principle this can be taken into account as

$$\langle n_{X,n,\mathrm{exp}}^\mathrm{coll}\rangle = \int_0^1 \mathrm{PDF}(\eta)\,\langle n_{X,n}^\mathrm{coll}(\eta)\rangle\,\mathrm{d}\eta + \left(1 - \int_0^1 \mathrm{PDF}(\eta)\,\mathrm{d}\eta\right)\langle n_{X,n}^\mathrm{q}\rangle, \tag{26}$$

where $\langle n_{X,n,\mathrm{exp}}^\mathrm{coll}\rangle$ is the experimentally expected number of photons in detector $X$ and mode $n$ including a stochastic state collapse, $\langle n_{X,n}^\mathrm{coll}(\eta)\rangle$ corresponds to the number of photons after collapse calculated in section 5 and $\langle n_{X,n}^\mathrm{q}\rangle$ is the number of photons expected from quantum evolution of the system as calculated in section 3. PDF $(\eta)$ is the probability density function for $\eta$ normalised to the probability that the collapse occurs in the interferometer. From these average photon numbers the visibility can be calculated using equation (10). In case of a number state collapse the contribution to the interference visibility after a collapse equals 0, see section 5, and the visibility will decrease according to the probability that the collapse occurs in the interferometer. On the other hand, for a coherent state collapse the visibility after collapse is unequal to 0 and thus we would need an explicit model for the stochasticity of the collapse process. Although we have not performed the calculation for a coherent state collapse, we may still expect the same behaviour as described before, i.e., as soon as the collapse process sets in the interference visibility decreases faster to $1/3$ than expected from our calculations presented in section 3, after which the visibility will decrease to $1/5$, while increasing the gain of both amplifiers further.

Under these considerations, an experiment with two $40\,\mathrm{dB}$ amplifiers ($\kappa = 4.7$) at $50\,\mathrm{mK}$, which might be developed if losses are reduced, is feasible.

# 7    Conclusions

We conclude that it should be possible to determine whether or not a $40\,\mathrm{dB}$-microwave parametric amplifier causes a wave function to collapse. If we insert such an amplifier into each of the two arms of an interferometer, we can measure the visibility of the output radiation. Neglecting losses the interference visibility of both signal and idler tend to $1/3$ with increasing gain, in case no collapse takes place. If the state collapses onto a number state within the interferometer, the visibility reduces to 0, whereas we found a significant deviation from $1/3$ in the case that the state collapses onto a coherent state. In case the insertion loss of the amplifiers is $2.2\,\mathrm{dB}$, while the temperature of the devices is $50\,\mathrm{mK}$, we estimate an interference visibility of 0.26 at large amplifier gain. In case wave function collapse sets in, we still expect the visibility to decrease measurably.

# Acknowledgements

We would like to thank M.J.A. de Dood for fruitful discussions and C.W.J. Beenakker for the use of the computer cluster. We thank M. de Wit for proofreading this manuscript. We also express our gratitude to the Frontiers of Nanoscience programme, supported by the Netherlands Organisation for Scientific Research (NWO/OCW), for financial support.

# A    Experimental realisation using resonator based parametric amplifiers

The discussed set-up is not the only conceivable realisation of the experiment. Instead of using a TWPA, it is also possible to use a resonator based parametric amplifier, such as the Josephson parametric amplifier (JPA), if the bandwidth of the photons is smaller than the bandwidth of the amplifier. TWPAs are broadband ($BW \approx 5\mathrm{GHz}$ [9]), whereas JPAs are intrinsically limited in their bandwidth ($BW \approx$

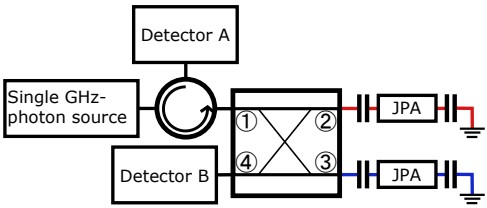

Figure 5: Schematic overview of the implementation of the experiment using JPAs. In this case it is beneficial to use a Michelson type interferometer to minimise losses.

10MHz [3]). However, both amplifiers are suitable to amplify a single photon with a $1\,\mathrm{MHz}$-bandwidth, in case our photon source would have a $T_1$-time in excess of $1\,\mu$s.

As we want to minimise losses and reflections in the interferometer arms, using a TWPA leads to a Mach-Zehnder type interferometer, whereas using a JPA results in a Michelson type interferometer, see figure 5. In case the JPA works in the non-degenerate regime ($\omega_{\mathrm{s}} \neq \omega_{\mathrm{i}}$), the results of the interference visibility as presented in this paper are the same.

# B  Non-degenerate vs. degenerate amplifiers

In the main text we considered the amplifiers to be non-degenerate, i.e. $\omega_{\mathrm{s}} \neq \omega_{\mathrm{i}}$. In case the amplifiers work in a degenerate regime,

$$\hat{H}_{\mathrm{deg}} = -\hbar\chi \left( \hat{a}_{\mathrm{s}}^{\dagger}\hat{a}_{\mathrm{s}}^{\dagger}\mathrm{e}^{i\Delta\phi} + \mathrm{H.c.} \right) \tag{27}$$

and the amplification will be dependent on the relative phase, $\Delta\phi$, between the signal and the pump, see figure 6. In this case we can still measure a visibility – in fact, $\Delta\phi$ can be used as a phase shifter in the experiment – as can be observed in figure 7. In this figure, the expected interference visibility in case the quantum state does not collapse within the interferometer is depicted using continuous lines. In case we assume that the state collapses into a noiseless coherent state in between the amplifiers and the second hybrid, the resulting visibility can be calculated using the method outlined in section 5 and appendix H. The result is depicted in figure 7 using dashed lines. It is observed that for large amplification $\kappa$ the two results approach each other asymptotically.

The main advantage of using non-degenerate instead of degenerate amplifiers is that the latter have not been developed. In the microwave regime, parametric amplifiers have been developed using Josephson junctions and kinetic inductance as the source of non-linear wave mixing and the resulting amplification. Both these sources lead naturally to non-degenerate devices as the non-linearity scales quadratically with pump current. One can use these as quasi-degenerate amplifiers by, e.g., biasing the device using a DC-current. This complicates the set-ups as proposed in figures 1 and 5, which can be a source of reflections and decoherence. Moreover, such amplifiers will always have non-degenerate contributions to their amplification, which complicates the analysis of the experiment. Thirdly, non-degenerate amplifiers enable one to study two interference visibilities (of both signal and idler) instead of one. For these reasons, we consider non-degenerate amplifiers to be more suited for our proposed experiment.

# C  Analytical model

Without losses and using the assumptions for the TWPAs as presented in section 3, we can obtain an analytical expression for the output state. We start by creating a single signal photon in input channel 1.

$$|\psi\rangle_1 = \hat{a}_{1\mathrm{s}}^{\dagger} |0_{1\mathrm{s}}, 0_{1\mathrm{i}}, 0_{4\mathrm{s}}, 0_{4\mathrm{s}}\rangle = |1_{1\mathrm{s}}, 0_{1\mathrm{i}}, 0_{4\mathrm{s}}, 0_{4\mathrm{s}}\rangle \tag{28}$$

Here, $\hat{a}^{\dagger}$ is the creation operator working on the vacuum. We then incorporate the $90^{\circ}$-hybrid by making the transformation

$$\hat{a}_{1\mathrm{s}}^{\dagger} \mapsto \frac{1}{\sqrt{2}} \left( i\hat{a}_{2\mathrm{s}}^{\dagger} + \hat{a}_{3\mathrm{s}}^{\dagger} \right). \tag{29}$$

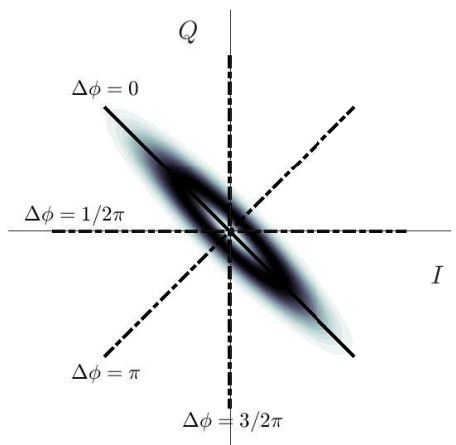

Figure 6: Wigner function of the state entering the hybrid after amplification by a degenerate amplifier (equation (27)). Depicted is the case where the signal and pump are in phase ($\Delta\phi = 0$). If $\Delta\phi \neq 0$ the Wigner function rotates according to the dash-dotted lines.

Next, a phase shift $\Delta\theta$ is applied to the upper arm,

$$\hat{a}_{2\mathrm{s}}^{\dagger} \mapsto \hat{a}_{2\mathrm{s}}^{\dagger} \mathrm{e}^{i\theta \hat{a}_{2\mathrm{s}}^{\dagger} \hat{a}_{2\mathrm{s}}} \tag{30}$$

at which the state just before the TWPAs is

$$|\psi\rangle_2 = \frac{1}{\sqrt{2}} \left( i\mathrm{e}^{i\Delta\theta \hat{a}_{2\mathrm{s}}^{\dagger} \hat{a}_{2\mathrm{s}}} \hat{a}_{2\mathrm{s}}^{\dagger} + \hat{a}_{3\mathrm{s}}^{\dagger} \right) |0_{2\mathrm{s}}, 0_{2\mathrm{i}}, 0_{3\mathrm{s}} 0_{3\mathrm{s}}\rangle \tag{31}$$

$$= \frac{1}{\sqrt{2}} \left( i\mathrm{e}^{i\Delta\theta} |1_{2\mathrm{s}}, 0_{2\mathrm{i}}, 0_{3\mathrm{s}} 0_{3\mathrm{s}}\rangle + |0_{2\mathrm{s}}, 0_{2\mathrm{i}}, 1_{3\mathrm{s}} 0_{3\mathrm{s}}\rangle \right). \tag{32}$$

For the TWPAs we use the following Hamiltonian in the interaction picture

$$\hat{H}_{\mathrm{TWPA}}^{\mathrm{eff}} = -\hbar\chi \left( \hat{a}_{\mathrm{s}}^{\dagger} \hat{a}_{\mathrm{i}}^{\dagger} + \hat{a}_{\mathrm{s}} \hat{a}_{\mathrm{i}} \right). \tag{33}$$

Evolving the state under this Hamiltonian as $|\psi\rangle_3 = \mathrm{e}^{-i\hat{H}_{\mathrm{TWPA}}^{\mathrm{eff}} t/\hbar}$, the output for a single amplifier in a single arm is (cf. [31])

$$\mathrm{e}^{-i\hat{H}_{\mathrm{TWPA}} t/\hbar} |N_{\mathrm{s}}, 0_{\mathrm{i}}\rangle = \cosh^{-(1+N_{\mathrm{s}})}\kappa \sum_{n=0}^{\infty} \frac{(i\tanh\kappa)^n}{n!} \left( \hat{a}_{\mathrm{s}}^{\dagger} \hat{a}_{\mathrm{i}}^{\dagger} \right)^n |N_{\mathrm{s}}, 0_{\mathrm{i}}\rangle \tag{34}$$

– or, in case of a degenerate amplifier

$$\mathrm{e}^{-i\hat{H}_{\mathrm{deg}} t/\hbar} |N_{\mathrm{s}}, 0_{\mathrm{i}}\rangle = \cosh^{-1/2(1+2N_{\mathrm{s}})} 2\kappa \sum_{n=0}^{\infty} \frac{\left( i\mathrm{e}^{i\Delta\phi}/2 \tanh 2\kappa \right)^n}{n!} \left( \hat{a}_{\mathrm{s}}^{\dagger} \hat{a}_{\mathrm{i}}^{\dagger} \right)^n |N_{\mathrm{s}}\rangle \ -, \tag{35}$$

where $N_{\mathrm{s}}$ is the number of signal photons initially present and $\kappa \equiv \chi t$. Applying this relation to $|\psi\rangle_2$, we obtain the state after the TWPAs.

$$|\psi\rangle_3 = \frac{1}{\sqrt{2}} \left[ \cosh^{-2}\kappa \cosh^{-1}\kappa' i\mathrm{e}^{i\Delta\theta} \sum_{n,m=0}^{\infty} \frac{i^n \tanh^n \kappa}{n!} \frac{i^m \tanh^m \kappa'}{m!} \left( \hat{a}_{5\mathrm{s}}^{\dagger} \hat{a}_{5\mathrm{i}}^{\dagger} \right)^n \left( \hat{a}_{8\mathrm{s}}^{\dagger} \hat{a}_{8\mathrm{i}}^{\dagger} \right)^m \hat{a}_{5\mathrm{s}}^{\dagger} + \right.$$

$$\left. + \cosh^{-1}\kappa \cosh^{-2}\kappa' \sum_{n,m=0}^{\infty} \frac{i^n \tanh^n \kappa}{n!} \frac{i^m \tanh^m \kappa'}{m!} \left( \hat{a}_{5\mathrm{s}}^{\dagger} \hat{a}_{5\mathrm{i}}^{\dagger} \right)^n \left( \hat{a}_{8\mathrm{s}}^{\dagger} \hat{a}_{8\mathrm{i}}^{\dagger} \right)^m \hat{a}_{8\mathrm{s}}^{\dagger} \right]. \tag{36}$$

$$\cdot |0_{5\mathrm{s}}, 0_{5\mathrm{i}}, 0_{8\mathrm{s}} 0_{8\mathrm{s}}\rangle,$$

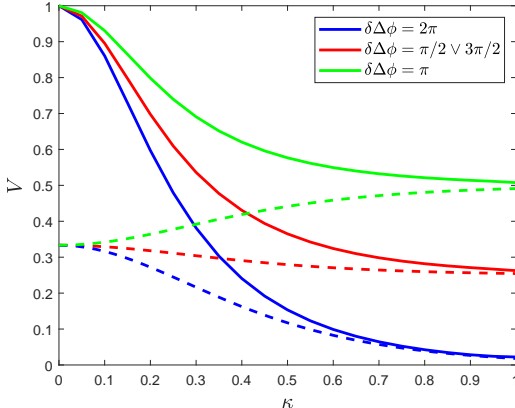

Figure 7: Interference visibility of the experiment implementing degenerate parametric amplifiers as function of amplification $\kappa = \chi \Delta t_{\text{deg}}$ and the difference in relative phase of the two amplifiers, $\delta \Delta \phi = \Delta \phi_{\text{up}} - \Delta \phi_{\text{low}}$. $\delta \Delta \phi$ can effectively be used as a phase shifter and we assume the interferometer to be lossless. The continous lines represent the visibility resulting from a quantum calculation. The dashed lines result from a calculation in which we assume state collapse into coherent states between the amplifiers and the second hybrid ($\eta = 1$, see section 5 and appendix H).

where $\kappa$ and $\kappa'$ are the amplification in the upper arm and lower arm respectively. Finally, the state traverses the second hybrid which is modelled by the transformations

$$
\begin{aligned}
\hat{a}_5^\dagger &\mapsto \frac{1}{\sqrt{2}} \left( i \hat{a}_6^\dagger + \hat{a}_7^\dagger \right) \\
\hat{a}_8^\dagger &\mapsto \frac{1}{\sqrt{2}} \left( \hat{a}_6^\dagger + i \hat{a}_7^\dagger \right)
\end{aligned}
\tag{37}
$$

for both signal and idler. Thus, we arrive at the output state

$$
\begin{aligned}
|\psi\rangle_4 =& \frac{1}{2} \cosh^{-1} \kappa \cosh^{-1} \kappa' \left[ \left( \frac{-e^{i\Delta\theta}}{\cosh\kappa} + \frac{1}{\cosh\kappa'} \right) \hat{a}_{6s}^\dagger + \left( \frac{ie^{i\Delta\theta}}{\cosh\kappa} + \frac{i}{\cosh\kappa'} \right) \hat{a}_{7s}^\dagger \right] \cdot \\
& \cdot \sum_{n,m=0}^\infty \frac{i^n \tanh^n \kappa}{2^n n!} \frac{i^m \tanh^m \kappa'}{2^m m!} \left( -\hat{a}_{6s}^\dagger \hat{a}_{6i}^\dagger + i \left\{ \hat{a}_{6s}^\dagger \hat{a}_{7i}^\dagger + \hat{a}_{7s}^\dagger \hat{a}_{6i}^\dagger \right\} + \hat{a}_{7s}^\dagger \hat{a}_{7i}^\dagger \right)^n \\
& \left( \hat{a}_{6s}^\dagger \hat{a}_{6i}^\dagger + i \left\{ \hat{a}_{6s}^\dagger \hat{a}_{7i}^\dagger + \hat{a}_{7s}^\dagger \hat{a}_{6i}^\dagger \right\} - \hat{a}_{7s}^\dagger \hat{a}_{7i}^\dagger \right)^m |0_{6s}, 0_{6i}, 0_{7s} 0_{7s}\rangle .
\end{aligned}
\tag{38}
$$

This equation reproduces the interference visibilities as presented in figure 2 in case losses are neglected.

# D    Output of numerical calculations

From our numerical calculations we obtain the probability distribution of number states, $P(\langle n \rangle_{\text{A,s}} = i, \langle n \rangle_{\text{A,i}} = j, \langle n \rangle_{\text{B,s}} = k, \langle n \rangle_{\text{B,i}} = l)$ in detectors A and B ($i, j, k, l \in [0, N-1]$). Using partial traces, we can compute the statistics and correlations for each of the four modes and between pairs of modes. E.g. the number state probability distribution for signal photons in detector B is depicted in figure 8.

In figure 9 we depict the photon number correlations between the input arms of the second hybrid (arms 5 (top) and 8 (bottom)) for amplifications $\kappa = 0$, 0.5 and 1. The top row in the figure ((a)-(c)) shows the correlations between the amount of signal photons in both arms. It can be observed that the correlations are symmetric around the line $n_{5s} = n_{8s}$. The second row ((d)-(f)) depicts the correlations between the number of signal and idler photons in arm 5. As can be seen, the number of idler photons is always equal to the number of signal photons or less by 1, as expected. The final row ((g)-(i)) shows the correlations between the number of idler photons in arms 5 and 8. For increased amplification these correlations look more and more like the correlations for the signal photons.

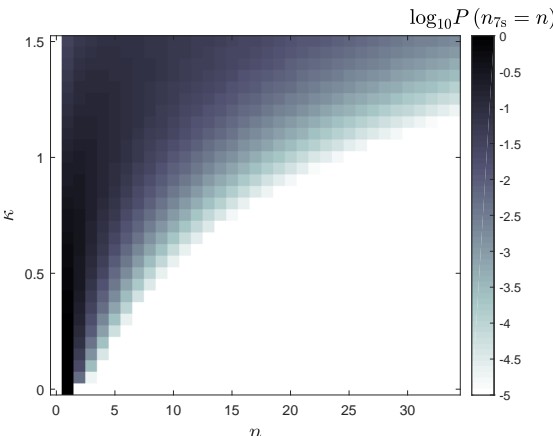

Figure 8: Probability distribution of the interferometer's output in arm 7 (detector B) for the signal mode as a function of amplification $\kappa$. The probabilities are cut-off at $P < 10^{-5}$.

Figure 9: Photon number correlations just before the second hybrid for various amplifications $\kappa$. (a)-(c) Correlations between number of signal photons in arms 5 and 8. (d)-(f) Correlations between the number of signal photons and idler photons in arm 5. (g)-(i) Correlations between the number of idler photons in arms 5 and 8. The colourbars are cut-off at $P < 10^{-5}$.

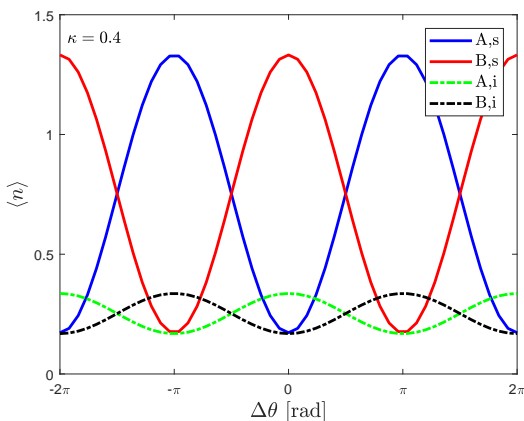

Figure 10: Predicted interference pattern of the interferometer in figure 1 (losses neglected): the average number of signal and idler photons in detectors A and B for amplification 0.4. At phase shift $\Delta\theta = 0$ most of the signal photons are expected in detector A, whereas most of the idler photons end up in detector B.

## E    Definition of interference visibility

In the main text the interference visibility is defined as

$$V_{\text{s(i)}} \equiv \left. \frac{\langle n_{\text{B,s(A,i)}} \rangle - \langle n_{\text{A,s(B,i)}} \rangle}{\langle n_{\text{B,s(A,i)}} \rangle + \langle n_{\text{A,s(B,i)}} \rangle} \right|_{\Delta\theta=0} . \tag{39}$$

The rationale behind this definition can be found in figure 10. At $\Delta\theta = 0$ we expect the maximum number of signal photons in detector B and the minimum in detector A. For the idler the opposite is the case.

## F    Comparison of full and reduced Hilbert space

As mentioned, the Hilbert space of the full interferometer scales as $N^4$ (no loss) and the number of entries in the density matrix scales as $N^8$ (with loss). However, if the amplifiers are identical, we can obtain the same result if we perform the calculation twice – once with a $|1\rangle_{\text{s}} |0\rangle_{\text{i}}$ input state and once with a $|0\rangle_{\text{s}} |0\rangle_{\text{i}}$ input state. The first yields $\langle n_{\text{B,s (A,i)}} \rangle$ and the second $\langle n_{\text{A,s (B,i)}} \rangle$. This implies that the same results can be obtained with a Hilbert space of $2N^2$ (no loss) or $2N^4$ (with loss).

In figure 11 the result of the two calculations is compared as a function of $\Gamma \Delta t_{\text{TWPA}}$ for $\kappa = 0.1$ to 0.4. In this figure, the grey solid data correspond to QUTIP's master equation solver, whereas the black dashed data are obtained using the reduced Hilbert space approach. As can be seen, the results overlap very well, such that we can use the reduced Hilbert space for our calculations.

## G    Interference visibility with losses

In case transmission losses are taken into account, we can fit the average number of photons leaving the interferometer with the function

$$\langle n_{\text{s(i)}} \rangle_{\text{out}} = \langle n_{\text{s(i)}} \rangle_{\text{out}} |_{\kappa=0} \cosh^2 \kappa + \left( \langle n_{\text{i(s)}} \rangle_{\text{out}} |_{\kappa=0} + 1 \right) \text{e}^{-f} \sinh^2 \kappa \tag{40}$$

in which $f$ is a fitting parameter depending on $\Gamma$, the various $\Delta t$s, $n_{\text{th}}$ and the input state.

$$\langle n \rangle_{\text{out}} |_{\kappa=0} = \left( \langle n \rangle_{\text{in}} - n_{\text{th}} \right) \text{e}^{-\Gamma \Delta t_{\text{tot}}} + n_{\text{th}} \tag{41}$$

is the number of photons leaving the interferometer in case the amplification $\kappa$ equals 0. The result of a particular fit ($\Gamma = 100\,\text{MHz}$, $\Delta t_{\text{TWPA}} = 10\,\text{ns}$ – other $\Delta t$s are $1\,\text{ns}$, hence $\Gamma \Delta t_{\text{tot}} = 1.3$ –, $n_{\text{th}} = 8.3 \times 10^{-3}$) is presented in figure 12. In figure 13 the magnitude of the fitting factor $f$ is plotted as a function of $\Gamma \Delta t_{\text{tot}}$ and $n_{\text{th}}$.

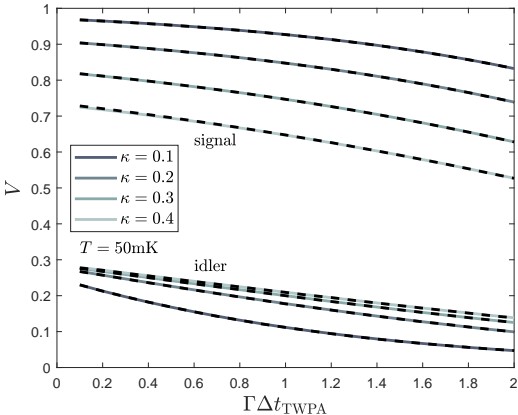

Figure 11: Visibility as a a function of losses in the TWPAs for various $\kappa$. $\Gamma = 100\,\text{MHz}$, $T = 50\,\text{mK}$, $\omega_{\text{s,i}} = 2\pi \times 5\,\text{GHz}$. $\Gamma\Delta t = 0.1$ in the other components of the set-up. The data in grey (solid) are obtained from QuTip's master equation solver using a $N^8$ Hilbert space with $N = 5$. Overlain (black dashed) are the data obtained from the reduced Hilbert space ($2N^4$, see text). As can be observed, the overlap is very good.

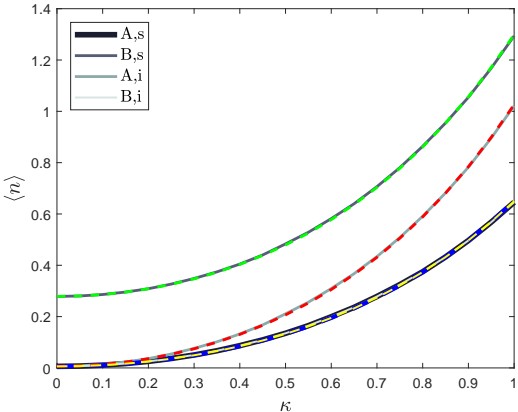

Figure 12: Average number of signal and idler photons reaching the detector as a function of $\kappa$ ($\Gamma = 100\,\text{MHz}$, $\Delta t_{\text{TWPA}} = 10\,\text{ns}$ – other $\Delta t$s are $1\,\text{ns}$, hence $\Gamma\Delta t_{\text{tot}} = 1.3$ –, $n_{\text{th}} = 8.3 \times 10^{-3}$). In grey the output from the reduced Hilbert space calculation. The coloured dashed lines are the result from a fit using equation (40). Note that the curves for signal photons in detector A and idler photons in detector B are overlapping.

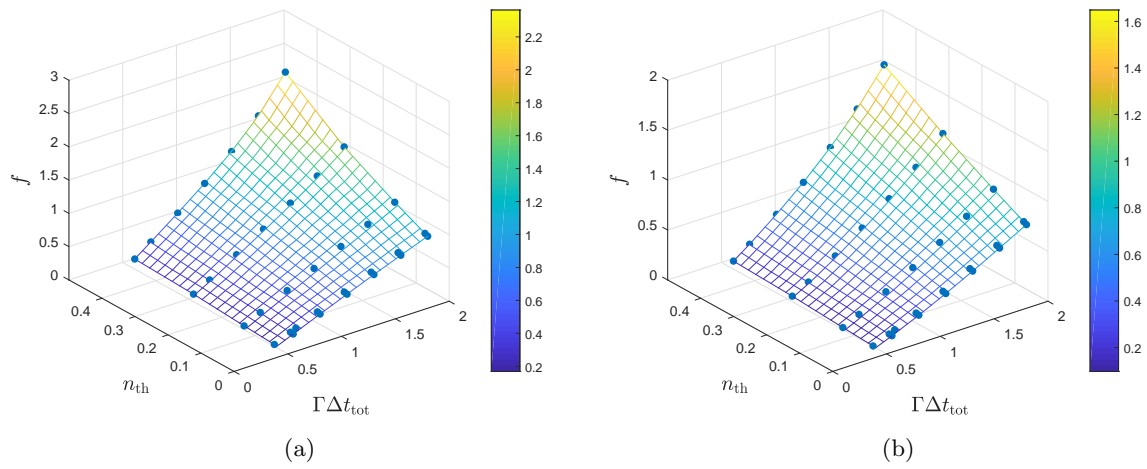

(a)                                                                        (b)

Figure 13: Magnitude of the fitting factor $f$ as function of $\Gamma\Delta t_{\text{tot}}$ and $n_{\text{th}}$ for the case $\Gamma = 100\,\text{MHz}$ and $\Delta t_{\text{h}_1,\text{ps},\text{h}_2} = 1\,\text{ns}$. (a) should be used for calculating (A,s), (B,s) and (B,i), whereas (b) should be used for (A,i). The dots represent the numerical data, whereas the mesh is a linear interpolation.

# H    Interference visibility with collapse onto coherent states

To study the interference visibility in case of state collapse within the interferometer, we assume that the state collapses into a coherent state, the most classical state available in quantum mechanics. Coherent states are expanded in Fock space as

$$|\alpha\rangle = e^{-|\alpha|^2/2} \sum_{n=0}^{\infty} \frac{\alpha^n}{\sqrt{n!}} |n\rangle \tag{42}$$

in which $\alpha \in \mathbb{C}$ is the amplitude of the coherent state and $|n\rangle$ are the number states. The mean number of photons in a coherent state equals $|\alpha|^2$. From equation (42) we can easily compute the overlap between a coherent state and a number state as

$$\langle\alpha|n\rangle = e^{-|\alpha|^2/2} \frac{(\alpha^*)^n}{\sqrt{n!}}. \tag{43}$$

Assuming that the interferometer is lossless and that the collapse takes place within the interferometer, the squared overlap between the collapsed coherent state $|\psi\rangle_{\mathrm{coll}} = |\alpha_{\mathrm{up,s}}\rangle |\alpha_{\mathrm{up,i}}\rangle |\alpha_{\mathrm{low,s}}\rangle |\alpha_{\mathrm{low,i}}\rangle$ and the instantaneous quantum state, given by equation (36) with $\kappa \mapsto \eta\kappa$, is

$$
\begin{aligned}
|c_{\mathrm{coll}}|^2 = |\langle\psi_{\mathrm{coll}}|\psi_3\rangle|^2 = &\frac{e^{-\left(|\alpha_{\mathrm{up,s}}|^2 + |\alpha_{\mathrm{up,i}}|^2 + |\alpha_{\mathrm{low,s}}|^2 + |\alpha_{\mathrm{low,i}}|^2\right)}}{2\cosh^6 \eta\kappa} \cdot \\
& \cdot \left(|\alpha_{\mathrm{up,s}}|^2 + |\alpha_{\mathrm{low,s}}|^2 + \left(i\,|\alpha_{\mathrm{up,s}}|\,|\alpha_{\mathrm{low,s}}|\,e^{i(\phi_{\mathrm{low,s}} - \phi_{\mathrm{up,s}})} + c.c.\right)\right) \cdot \\
& \cdot \sum_{n,m,l,k} \frac{(i)^{n+m-l-k} \tanh^{n+m+l+k} \eta\kappa}{n!m!l!k!} (|\alpha_{\mathrm{up,s}}|\,|\alpha_{\mathrm{up,i}}|)^{n+l} (|\alpha_{\mathrm{low,s}}|\,|\alpha_{\mathrm{low,i}}|)^{m+k} \cdot \\
& \cdot e^{i(n-l)(\phi_{\mathrm{up,s}} + \phi_{\mathrm{up,i}}) + (m-k)(\phi_{\mathrm{low,s}} + \phi_{\mathrm{low,i}})}
\end{aligned}
\tag{44}
$$

in case the amplifiers are equal and setting the amplitudes to $\alpha = |\alpha|\,e^{i\phi_\alpha}$. The amplifiers evolve the amplitudes of the collapsed state $|\psi_{\mathrm{coll}}\rangle$ further into average amplitudes

$$\bar{\alpha}_{\mathrm{up(low),s(i)}} = \alpha_{\mathrm{up(low),s(i)}} \cosh(1-\eta)\kappa + i\alpha^*_{\mathrm{up(low),i(s)}} \sinh(1-\eta)\kappa \tag{45}$$

and the number of photons arriving in each of the detectors for this particular collapse equals

$$n^{\mathrm{coll}}_{[A]\{B\},n} = \frac{1}{2} |[i]\{1\}\bar{\alpha}_{\mathrm{up},n} + [1]\{i\}\bar{\alpha}_{\mathrm{low},n}|^2. \tag{46}$$

In the last expression we have used the standard hybrid transformation relations

$$\alpha_{[A]\{B\},n} = \frac{1}{\sqrt{2}} \left([i]\{1\}\alpha_{\mathrm{up},n} + [1]\{i\}\alpha_{\mathrm{low},n}\right) \tag{47}$$

as well as that $n^{\mathrm{coll}}_{\mathrm{A(B)},n} = |\alpha_{\mathrm{A(B)},n}|^2$. Explicitly,

$$
\begin{aligned}
n^{\mathrm{coll}}_{[A]\{B\},s} = \frac{1}{2}\Bigg[ &\left(|\alpha_{\mathrm{up,s}}|^2 + |\alpha_{\mathrm{low,s}}|^2\right)\cosh^2(1-\eta)\kappa + \left(|\alpha_{\mathrm{up,i}}|^2 + |\alpha_{\mathrm{low,i}}|^2\right)\sinh^2(1-\eta)\kappa - \\
& - \left(i\,|\alpha_{\mathrm{up,s}}|\,|\alpha_{\mathrm{up,i}}|\,e^{i(\phi_{\mathrm{up,s}} + \phi_{\mathrm{up,i}})}\cosh(1-\eta)\kappa\sinh(1-\eta)\kappa + c.c.\right) + \\
& + [1]\{-1\}\left(i\,|\alpha_{\mathrm{up,s}}|\,|\alpha_{\mathrm{low,s}}|\,e^{i(\phi_{\mathrm{up,s}} - \phi_{\mathrm{low,s}})}\cosh^2(1-\eta)\kappa + c.c.\right) + \\
& + [1]\{-1\}\left(|\alpha_{\mathrm{up,s}}|\,|\alpha_{\mathrm{low,i}}|\,e^{i(\phi_{\mathrm{up,s}} + \phi_{\mathrm{low,i}})}\cosh(1-\eta)\kappa\sinh(1-\eta)\kappa + c.c.\right) + \\
& + [-1]\{1\}\left(|\alpha_{\mathrm{up,i}}|\,|\alpha_{\mathrm{low,s}}|\,e^{-i(\phi_{\mathrm{up,i}} + \phi_{\mathrm{low,s}})}\cosh(1-\eta)\kappa\sinh(1-\eta)\kappa + c.c.\right) + \\
& + [1]\{-1\}\left(i\,|\alpha_{\mathrm{up,i}}|\,|\alpha_{\mathrm{low,i}}|\,e^{-i(\phi_{\mathrm{up,i}} - \phi_{\mathrm{low,i}})}\sinh^2(1-\eta)\kappa + c.c.\right) - \\
& - \left(i\,|\alpha_{\mathrm{low,s}}|\,|\alpha_{\mathrm{low,i}}|\,e^{i(\phi_{\mathrm{low,s}} + \phi_{\mathrm{low,i}})}\cosh(1-\eta)\kappa\sinh(1-\eta)\kappa + c.c.\right)\Bigg],
\end{aligned}
\tag{48}
$$

$$n_{[A]\{B\},i}^{\text{coll}} = \frac{1}{2}\Bigg[ \left(|\alpha_{\text{up,s}}|^2 + |\alpha_{\text{low,s}}|^2\right)\sinh^2\left(1-\eta\right)\kappa + \left(|\alpha_{\text{up,i}}|^2 + |\alpha_{\text{low,i}}|^2\right)\cosh^2\left(1-\eta\right)\kappa - $$

$$- \left(i\,|\alpha_{\text{up,s}}|\,|\alpha_{\text{up,i}}|\,e^{i(\phi_{\text{up,s}}+\phi_{\text{up,i}})}\sinh\left(1-\eta\right)\kappa\cosh\left(1-\eta\right)\kappa + c.c.\right) + $$

$$+ [1]\{-1\}\left(i\,|\alpha_{\text{up,s}}|\,|\alpha_{\text{low,s}}|\,e^{-i(\phi_{\text{up,s}}-\phi_{\text{low,s}})}\sinh^2\left(1-\eta\right)\kappa + c.c.\right) + $$

$$+ [-1]\{1\}\left(|\alpha_{\text{up,s}}|\,|\alpha_{\text{low,i}}|\,e^{-i(\phi_{\text{up,s}}+\phi_{\text{low,i}})}\sinh\left(1-\eta\right)\kappa\cosh\left(1-\eta\right)\kappa + c.c.\right) + \quad (49)$$

$$+ [1]\{-1\}\left(|\alpha_{\text{up,i}}|\,|\alpha_{\text{low,s}}|\,e^{i(\phi_{\text{up,i}}+\phi_{\text{low,s}})}\sinh\left(1-\eta\right)\kappa\cosh\left(1-\eta\right)\kappa + c.c.\right) + $$

$$+ [1]\{-1\}\left(i\,|\alpha_{\text{up,i}}|\,|\alpha_{\text{low,i}}|\,e^{i(\phi_{\text{up,i}}-\phi_{\text{low,i}})}\cosh^2\left[(1-\eta)\,\kappa\right] + c.c.\right) - $$

$$- \left(i\,|\alpha_{\text{low,s}}|\,|\alpha_{\text{low,i}}|\,e^{i(\phi_{\text{low,s}}+\phi_{\text{low,i}})}\sinh\left(1-\eta\right)\kappa\cosh\left(1-\eta\right)\kappa + c.c.\right)\Bigg].$$

With these ingredients we can obtain the average number of photons arriving in each of the detectors as

$$\langle n_{X,n}^{\text{coll}}\rangle = \frac{1}{\pi^4}\int n_{X,n}^{\text{coll}}\,|c_{\text{coll}}|^2\,\mathrm{d}^2\alpha_{\text{up,s}}\,\mathrm{d}^2\alpha_{\text{up,i}}\,\mathrm{d}^2\alpha_{\text{low,s}}\,\mathrm{d}^2\alpha_{\text{low,i}} \quad (50)$$

as discussed in the main text. Here, $\mathrm{d}^2\alpha = |\alpha|\,\mathrm{d}\phi_\alpha\mathrm{d}\alpha$ and the bounds of the integrals are $[0,\infty\rangle$ for integration over the amplitudes and $[0,2\pi\rangle$ for integration over the phases.

Due to the complex exponentials in equations (44) and (48) and the integration over the full domain $[0,2\pi\rangle$ for the phases, it is immediately observed that the integrand of equation (50) only contributes to the integral for integrand terms that are independent of $\phi_{\text{up(low),s(i)}}$. Then, integration over the phases yields a factor $16\pi^4$.

For the calculation of $\langle n_{\text{B,s}}^{\text{coll}}\rangle - \langle n_{\text{A,s}}^{\text{coll}}\rangle$ and $\langle n_{\text{A,i}}^{\text{coll}}\rangle - \langle n_{\text{B,i}}^{\text{coll}}\rangle$ we find that only the terms scaling as $e^{\pm i(\phi_{\text{up,s}}-\phi_{\text{low,s}})}$ and $e^{\pm i(\phi_{\text{up,i}}-\phi_{\text{low,i}})}$ from equations (48) and (49) will contribute to the integral. For the term scaling as $e^{i(\phi_{\text{up,s}}-\phi_{\text{low,s}})}$ we find a contribution to $\langle n_{\text{B,s}}^{\text{coll}}\rangle - \langle n_{\text{A,s}}^{\text{coll}}\rangle$

$$\Delta_{\text{s,1}} = \frac{8\cosh^2\left(1-\eta\right)\kappa}{\cosh^6\eta\kappa}\int |\alpha_{\text{up,s}}|^3\,|\alpha_{\text{up,i}}|\,|\alpha_{\text{low,s}}|^3\,|\alpha_{\text{low,i}}|\,e^{-\left(|\alpha_{\text{up,s}}|^2+|\alpha_{\text{up,i}}|^2+|\alpha_{\text{low,s}}|^2+|\alpha_{\text{low,i}}|^2\right)}\cdot$$
$$\cdot B_0\left(2\,|\alpha_{\text{up,s}}|\,|\alpha_{\text{up,i}}|\tanh\eta\kappa\right)B_0\left(2\,|\alpha_{\text{low,s}}|\,|\alpha_{\text{low,i}}|\tanh\eta\kappa\right)\cdot \quad (51)$$
$$\cdot \mathrm{d}\,|\alpha_{\text{up,s}}|\,\mathrm{d}\,|\alpha_{\text{up,i}}|\,\mathrm{d}\,|\alpha_{\text{low,s}}|\,\mathrm{d}\,|\alpha_{\text{low,i}}|,$$

where we have used the identity $\sum_{n=0}^{\infty}x^{2n}/(n!)^2 = B_0(2x)$, in which $B_n(x)$ is the modified Bessel function of the first kind. For the contribution from equation (48) scaling as $e^{-i(\phi_{\text{up,s}}-\phi_{\text{low,s}})}$ we find the same expression. For the term in equation (48) scaling as $e^{i(\phi_{\text{up,i}}-\phi_{\text{low,i}})}$ we find a contribution

$$\Delta_{\text{s,2}} = \frac{8\sinh^2\left(1-\eta\right)\kappa}{\cosh^6\eta\kappa}\int |\alpha_{\text{up,s}}|^2|\alpha_{\text{up,i}}|^2|\alpha_{\text{low,s}}|^2|\alpha_{\text{low,i}}|^2 e^{-\left(|\alpha_{\text{up,s}}|^2+|\alpha_{\text{up,i}}|^2+|\alpha_{\text{low,s}}|^2+|\alpha_{\text{low,i}}|^2\right)}\cdot$$
$$\cdot \left[B_1\left(2\,|\alpha_{\text{up,s}}|\,|\alpha_{\text{up,i}}|\tanh\eta\kappa\right) - |\alpha_{\text{up,s}}|\,|\alpha_{\text{up,i}}|\tanh\eta\kappa\right]\cdot \quad (52)$$
$$\cdot \left[B_1\left(2\,|\alpha_{\text{low,s}}|\,|\alpha_{\text{low,i}}|\tanh\eta\kappa\right) - |\alpha_{\text{low,s}}|\,|\alpha_{\text{low,i}}|\tanh\eta\kappa\right]\cdot$$
$$\cdot \mathrm{d}\,|\alpha_{\text{up,s}}|\,\mathrm{d}\,|\alpha_{\text{up,i}}|\,\mathrm{d}\,|\alpha_{\text{low,s}}|\,\mathrm{d}\,|\alpha_{\text{low,i}}|$$

to $\langle n_{\text{B,s}}^{\text{coll}}\rangle - \langle n_{\text{A,s}}^{\text{coll}}\rangle$. Here we have used the identity $\sum_{n=0}^{\infty}x^{2n+1}/[(n+1)\,(n!)^2] = B_1(2x) - x$. Again, the contribution of the term in equation (48) scaling as $e^{-i(\phi_{\text{up,i}}-\phi_{\text{low,i}})}$ yields an equal contrbution, such that

$$\langle n_{\text{B,s}}^{\text{coll}}\rangle - \langle n_{\text{A,s}}^{\text{coll}}\rangle = 2\left(\Delta_{\text{s,1}} + \Delta_{\text{s,2}}\right). \quad (53)$$

For $\langle n_{\text{A,i}}^{\text{coll}}\rangle - \langle n_{\text{B,i}}^{\text{coll}}\rangle$ we find the similar expression

$$\langle n_{\text{A,i}}^{\text{coll}}\rangle - \langle n_{\text{B,i}}^{\text{coll}}\rangle = 2\left(\Delta_{\text{i,1}} + \Delta_{\text{i,2}}\right), \quad (54)$$

in which $\Delta_{\text{i,1(2)}}$ follow from equations (51) and (52) by replacing $\cosh\left(1-\eta\right)\kappa$ with $\sinh\left(1-\eta\right)\kappa$ and vice versa.

Similarly, we find that for the calculation of $\langle n_{\mathrm{B,s}}^{\mathrm{coll}}\rangle + \langle n_{\mathrm{A,s}}^{\mathrm{coll}}\rangle$ and $\langle n_{\mathrm{A,i}}^{\mathrm{coll}}\rangle + \langle n_{\mathrm{B,i}}^{\mathrm{coll}}\rangle$ only the terms without exponential factor and the terms scaling as $\mathrm{e}^{\pm i(\phi_{\mathrm{up,s}}+\phi_{\mathrm{up,i}})}$ and $\mathrm{e}^{\pm i(\phi_{\mathrm{low,s}}+\phi_{\mathrm{low,i}})}$ from equations (48) and (49) will contribute to the integral. For the terms without exponential we find a contribution

$$
\begin{aligned}
\Sigma_{\mathrm{s,1}} = \frac{8}{\cosh^6 \eta\kappa} \int & |\alpha_{\mathrm{up,s}}|\,|\alpha_{\mathrm{up,i}}|\,|\alpha_{\mathrm{low,s}}|\,|\alpha_{\mathrm{low,i}}| \left[\left(|\alpha_{\mathrm{up,s}}|^2 + |\alpha_{\mathrm{low,s}}|^2\right)\cosh^2(1-\eta)\,\kappa + \right. \\
& \left. + \left(|\alpha_{\mathrm{up,i}}|^2 + |\alpha_{\mathrm{low,i}}|^2\right)\sinh^2(1-\eta)\,\kappa\right] \cdot \\
& \cdot \left(|\alpha_{\mathrm{up,s}}|^2 + |\alpha_{\mathrm{low,s}}|^2\right)\mathrm{e}^{-\left(|\alpha_{\mathrm{up,s}}|^2 + |\alpha_{\mathrm{up,i}}|^2 + |\alpha_{\mathrm{low,s}}|^2 + |\alpha_{\mathrm{low,i}}|^2\right)} \cdot \\
& \cdot B_0\left(2\,|\alpha_{\mathrm{up,s}}|\,|\alpha_{\mathrm{up,i}}|\tanh\eta\kappa\right) B_0\left(2\,|\alpha_{\mathrm{low,s}}|\,|\alpha_{\mathrm{low,i}}|\tanh\eta\kappa\right) \cdot \\
& \cdot \mathrm{d}\,|\alpha_{\mathrm{up,s}}|\,\mathrm{d}\,|\alpha_{\mathrm{up,i}}|\,\mathrm{d}\,|\alpha_{\mathrm{low,s}}|\,\mathrm{d}\,|\alpha_{\mathrm{low,i}}|
\end{aligned} \tag{55}
$$

to $\langle n_{\mathrm{B,s}}^{\mathrm{coll}}\rangle + \langle n_{\mathrm{A,s}}^{\mathrm{coll}}\rangle$. Again, the contribution to $\langle n_{\mathrm{A,i}}^{\mathrm{coll}}\rangle + \langle n_{\mathrm{B,i}}^{\mathrm{coll}}\rangle$, $\Sigma_{\mathrm{i,1}}$, is the same except that $\cosh(1-\eta)\,\kappa \mapsto \sinh(1-\eta)\,\kappa$. For the term scaling as $\mathrm{e}^{i(\phi_{\mathrm{up,s}}+\phi_{\mathrm{up,i}})}$ we find a contribution

$$
\begin{aligned}
\Sigma_2 = \frac{8\cosh(1-\eta)\,\kappa\sinh(1-\eta)\,\kappa}{\cosh^6\eta\kappa} \int & |\alpha_{\mathrm{up,s}}|^2\,|\alpha_{\mathrm{up,i}}|^2\,|\alpha_{\mathrm{low,s}}|\,|\alpha_{\mathrm{low,i}}|\left(|\alpha_{\mathrm{up,s}}|^2 + |\alpha_{\mathrm{low,s}}|^2\right) \cdot \\
& \cdot \mathrm{e}^{-\left(|\alpha_{\mathrm{up,s}}|^2 + |\alpha_{\mathrm{up,i}}|^2 + |\alpha_{\mathrm{low,s}}|^2 + |\alpha_{\mathrm{low,i}}|^2\right)} \cdot \\
& \cdot \left[B_1\left(2\,|\alpha_{\mathrm{up,s}}|\,|\alpha_{\mathrm{up,i}}|\tanh\eta\kappa\right) - |\alpha_{\mathrm{up,s}}|\,|\alpha_{\mathrm{up,i}}|\tanh\eta\kappa\right] \cdot \\
& \cdot B_0\left(2\,|\alpha_{\mathrm{low,s}}|\,|\alpha_{\mathrm{low,i}}|\tanh\eta\kappa\right) \cdot \\
& \cdot \mathrm{d}\,|\alpha_{\mathrm{up,s}}|\,\mathrm{d}\,|\alpha_{\mathrm{up,i}}|\,\mathrm{d}\,|\alpha_{\mathrm{low,s}}|\,\mathrm{d}\,|\alpha_{\mathrm{low,i}}|
\end{aligned} \tag{56}
$$

to $\langle n_{\mathrm{B,s}}^{\mathrm{coll}}\rangle + \langle n_{\mathrm{A,s}}^{\mathrm{coll}}\rangle$ and $\langle n_{\mathrm{A,i}}^{\mathrm{coll}}\rangle + \langle n_{\mathrm{B,i}}^{\mathrm{coll}}\rangle$. The contribution from the other exponentially scaling terms from equations (48) and (49) contributing to the integral yield the same values, whence

$$
\langle n_{\mathrm{B,s}}^{\mathrm{coll}}\rangle + \langle n_{\mathrm{A,s}}^{\mathrm{coll}}\rangle = \Sigma_{\mathrm{s,1}} + 4\Sigma_2, \tag{57}
$$

$$
\langle n_{\mathrm{A,i}}^{\mathrm{coll}}\rangle + \langle n_{\mathrm{B,i}}^{\mathrm{coll}}\rangle = \Sigma_{\mathrm{i,1}} + 4\Sigma_2. \tag{58}
$$

Using equations (53), (57), (54) and (58) we easily compute the interference visibilities for signal and idler. We evaluated the integrals in these equations using MATHEMATICA.

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
