# Peer review of "An experimental proposal to study collapse of the wave function in travelling-wave parametric amplifiers"

_SciPost Physics_

## Round 3 · Referee Report · Anonymous · 2019-9-17

Strengths

The authors propose a novel model for spontaneous collapse in microwave quantum optical devices.

Weaknesses

The substantive part of the paper is based on their own model 'spontaneous collapse'. This is speculative and rather vague.

Report

I still think the calculation is trivial but, as this is a subjective judgement, I accept the authors response.

The authors respond that “Optical parametric amplifiers and microwave paramps work in a totally different regime of the EM-spectrum.” This is obviously true but irrelevant as the theoretical description is the same. In the case of super conducting circuits the quantum theory is given as an ‘effective theory’ in which particular macroscopic degrees of freedom (circuit variables) are quantised directly. This works well in typical experimental conditions because the underlying microscopic degrees of freedom largely factor-out only remaining as a possible source of decoherence or noise. This is an old argument that goes back to the seminal work of Leggett. Exactly the same approach is adopted in the quantisation of the non-linear polarizability of optically-active crystals and it works well there for the same reason. I simply do not understand the author’s comments about localised versus unlocalised electrons. The polarisation is a non-local effective degree of freedom. I thus disagree with the sentence in the new version of the paper that, “In optical non-linear crystals, the photons typically interact with localised electrons”: one can just as well claim that the photons interact with a continuously distributed polarisation field in the medium. I would strongly advise the authors to reconsider these claims.
In their response the authors say that, “ On the other hand, in microwave PAs the valence electrons (Cooper pairs) move freely through the material and the passage of a single photon will superpose them over a distance in the order of the photon wavelength (approximately 3mm)”. This suggests to me that they have in mind a different theoretical treatment to the usual one used in circuit QED, one that explicitly uses the microscopic theory of microwave electronics. I admit that such an approach might justify their phenomenological model but I would like to see how it is done. In any case I give the authors the benefit of the doubt. In a similar context the authors claim that “Collapse to a coherent state might take place, because the electrons in the transmission lines collapse into position states (classical motion can always be described by Fourier components, i.e. sinusoids, i.e. coherent states which are the closest quantum analogue to sinusoids).” I can’t make any sense of this. Electrons in transmission lines are effective field degrees of freedom.

In summary, I remain unconvinced by the author’s responses. But others may take a different view so I am prepared to recommend publication so that debate can take place.

---

## Round 3 · Referee Report · Anonymous · 2019-10-10

Report

The paper presents an interesting idea regarding amplification using travelling wave parametric amplifiers (TWPA), a standard technology for readout in quantum superconducting information processing. The idea is that the amplified superposition might spontaneously collapse because of the number of particles involved in the amplifier. However, the idea is not worked out in a rigorous way that puts it in the context of previous research so I cannot recommend publication in this, or any other, journal.

The work seems to be presented as a type of spontaneous collapse models. These should be universal, or at least broad, theories, precisely formulated, with suggested parameters designed to prevent the formation of macroscopic superpositions. By contrast, the present proposals apply only to the context of TWPAs, and are unsatisfactory in other ways. The first (number collapse) is so gross in its effects as to be surely wrong, while the second (coherent state collapse) has an ad hoc parameter so that it would not be clear how to apply it even for the slightly more general case of two TWPAs in series. It is also not clear when the authors imagine the model applies. I see two possibilities:

1) The evolution is postulated to be a property of TWPAs. If so, this simply means that TWPAs do not work according to the standard model of TWPAs, but rather have extra decoherence. But if that is the case, then why is the standard model considered accurate? In fact the authors say of TWPAs (p. 2) that “their quantum behaviour is well-understood”. The deviations should be testable for input states other than the single photon. Indeed, the models would surely have consequences for the very many published experiments using TWPAs, even with the gain the authors suggest as they say. The authors seem to ignore this issue.

2) The evolution only applies to the single-photon input. This makes the proposal even more ad hoc. Presumably the authors have some idea that the spontaneous collapse would prevent macroscopic entanglement. But they do not even show that there is any macroscopic entanglement when the practical amounts of noise in these amplifiers is included anyway. If the authors want a “no macroscopic entanglement” theory or a “no macroscopic superposition” theory then they need to make the theory sensitive to the presence of these things. The existence of fringes does not imply entanglement! For instance, a HOM dip in quantum optics can be observed, with visibility up to 50%, can be obtained from classical (mixtures of coherent states) fields.

Given the above criticisms, it seems to me that a better direction for the authors to take their work would be that of testing the existence of macroscopic superpositions, which they touch on in Section 6. However I am a bit skeptical of how impressive the comparison with molecular interferometers is. The authors say that 2 times 10^{-25} is “similar” to 7 times 10^{-24}, but is more than an order of magnitude smaller. As for “degrees of freedom” if that is to be the basis for a more favourable comparison then it would need to be fleshed out more. There is a considerable literature on measures of macroscopicity that the authors would need to engage with. They say (p. 2) that they are investigating an unexplored “part of parameter space” but they don’t say what the parameters are.

Requested changes

Finally, there are some small issues the authors should also address if they submit elsewhere.
i) The definition of gain below Eq.(3) does not look standard to me. It should be defined in terms of amplitudes, not intensities.
ii) The first, undidplayed, equation in Sec. 3 does not make sense. The arguments of P should be Fock numbers (integers), not mean values of Fock numbers.
iii) Even with loss as described in Sec. 4 the TWPA dynamics is linear with Gaussian noise. numerical simulation in Hilbert space is not necessary. There exist very efficient methods for propagating Gaussian states with such dynamics. Only a few parameters are needed.
iii) At the end of Sec. 5 the visibility figures of 0.15 and 0.2 is given. Are these exact or just approximations to the results of numerical simulations.

---

## Round 3 · Author Response

Dear Editor,
Thank you for the referee reports.

We thank the referees for their detailed reading of our manuscript and for their comments. Below we address their comments one by one and we indicate the changes to our manuscript in order to clarify the points they have raised.

Yours sincerely,
Tom van der Reep and Tjerk Oosterkamp

---

## Round 3 · List of Changes

Report 1
1 - While I neither verified the calculations explicitly nor have any specific reason to doubt them, I thought the paper would be greatly strengthened if some physical arguments were given to explain some of the interesting results.
(a) Some discussion of the strength of the correlations between photon number after the amplifier and before would have been welcome, and presumably one could proceed from this "distinguishability" to explain why the visibility reaches 1/3 in the high-gain regime.

TvdR: We have added such discussion in section 3, starting with the sentence 'Therefore, the visibility in the lossless case...'.
To add some intuition to the calculation we have added a figure to appendix D, to emphasize the point that a superposition of zero and one photon before an amplifier with gain G does not result in a superposition of zero and G photons after the amplifier. The visibility results from a calculation involving multiphoton interference involving many Fock states, which is not so straightforward.

(b) Although it is easier to see, the authors might have helped the readers' intuition by explaining why the signal visibility goes to 100% at low gain while the idler visibility goes to 0.

TvdR: For low gain the signal visibility goes to 1, as this situation resembles a 'normal' one photon interference experiment. The idler visibility goes to 0, due to a lack of idler photons. As a matter of fact, without losses the idler visibility is undefined. We have added a sentence ‘The low gain visibility of 1 for the signal.

(c) Discussion of the fact that even the idler visibility goes to 1/3 in the high-gain regime would help the reader develop a physical sense of the transition to classical-like correlations in this limit.

TvdR: Here the same mechanism is at work as in situation 1(a), discussed above. In the additional discussion in Sec. 3 we also mentioned the idler.

(d) In the discussion of loss as well, some arguments other than numerical could be provided to suggest how much loss would be necessary to decohere the states resulting from |0> and |1> as input. Most likely a consideration of <n_out> and \Delta n for the two cases, and what fraction of the output would need to be siphoned off in order to distinguish them, would explain this.

TvdR: We have added a paragraph in section 4 with analytical simplifications of the numerical interference visibility and discussed the limits of low temperature and low losses.

2 - The above issues are closely related to the earlier work prompted by de Martini et al about "micro / macro" entanglement using optical parametric amplifiers, where it was also observed that the coherence of the allegedly Schrödinger-cat-like state was surprisingly insensitive to loss. That prompted work by Gisin, Zeilinger, Simon, and others, which should really be discussed and cited in the present paper.

TvdR: We have added the references along with a short discussion. An important difference is that in a microwave version of those experiments losses are expected to be significantly higher in our case. It is the price we pay for

3 - The independence from loss in the high-gain limit suggests that one is essentially observing interference of classically correlated beams, rather than a true nonlocal quantum superposition. This might accord with some intuitions about high-gain OPAs (and the fact mentioned earlier that even the idlers exhibit good interference in this limit). I would have liked to see some discussion of whether or not the authors expected collapse into coherent states to degrade the visibility (or whether it would, in the high-gain limit, simply "confirm" and lock in the pre-existing phase correlations).

TvdR: Coming to think of it, indeed one might expect that a collapse to coherent states locks into the pre-existing phase correlations and ergo that the visibility does not change at all: We treat the coherent states as a point in phase space, whence our calculation might look as if we are taking the phase space of psi3 and time-develop each point separately through the second hybrid. Surprisingly, then, the visibilities are different, which is due to the addition of one photon in our collapse mechanism. In our manuscript we added some commenting lines on this matter.

3b - The dependence of visiblity on location of collapse is very interesting. But it requires some discussion, connected to point 3 above. Specifically: the fact that 1/3 visibility is preserved if the collapse occurs after the output of the high-gain amplifier is consistent with the idea that the phases are already correlated, and making those correlations "classical" changes nothing. But once one has these classically correlated phases, inserting two additional amplifiers should not, so far as I can tell, destroy the interference. But what is the difference between this situation and the eta=1/2 case (collapse halfway through the amplifiers), which the authors claim leads to 0 visibility? I worry that there is some error here -- if there is not, then it's certainly a point which merits some actual discussion in the text.

TvdR: Yes, there was an error and it has been fixed. As a result the corresponding figure changes: The visibility for eta=1 does not change. For eta=1/2 the visibility goes to approximately 0.15 in the high-gain limit. For eta=0 the visibility goes to 0.2 in the high-gain limit (see also next remark). Indeed the signal phases are correlated, see our result for |c_coll|^2 - it is more likely for the collapse to end up in a state with \phi_ups-\phi_lows=pi/2 than -pi/2. However, such a phase correlation is absent for the idlers, which implies that further transmission through the TWPA changes the correlations of signal and idler again (NB alpha_s(i)=alpha_s(i)0\cosh\kappa+i\alpha*_i(s)0\sinh\kappa).

3c - It's perhaps even more surprising that if collapse at the midpoint destroys interference entirely, collapse before the amplifers does not. I would expect discussion of this point. I would also think that a simple model could be produced to explain the 20% result when the collapse occurs upon the |01>+|10> state before the amplifiers.

TvdR: first point, see 3b - this is an error. Using the reduced Hilbert space approach one can easily explain the reduction in interference visibility for the cases \eta=0 and 1. Our collapse mechanism adds one photon in each Hilbert subspace (signal and idler, upper and lower arm). From this added photon the reduction can be explained. We have extended the paragraph on this matter in the manuscript.

4 - At the very beginning, the text could be clearer about the experimental situation. For instance, signal, idler, and pump could be defined earlier, and it should be made clear that in figure 1 it is the signal which is initially split, but that both the signal and the idler are to be interfered.

TvdR: Thank you, this has been adapted.

5 - In the next-to-last paragraph of section III, the sentence that "feeding this TWPA with a |00> state gives the average number of signal photons" doesn't make much sense to me. (The previous sentence, while awkward, at least makes it clear that what is meant -- I think -- is that equation 3 yields the number of signal photons if |10> is fed in; but I don't know what is supposed to yield that number if |00> is fed in.)

TvdR: the main idea of the reduced Hilbert space is that our interference visibility results without and with loss can be obtained by considering a single TWPA. For calculating the visibilities, four numbers are required: <n_As>, <n_Bs>, <n_Ai> and <n_Bi>. Now, instead of considering the full interferometer, one can use the reduced Hilbert space approach: If a single TWPA is fed with a |1_s,0_i>-state, the number of output photons (signal and idler) of this single TWPA equals <n_Bs> and <n_Ai> respectively. Feeding the single TWPA with a |1_s,0_i>-state, the output numbers of photons of this single TWPA are found to be equal to <n_As> and <n_Bi> respectively. This implies that if one performs these two calculations (with single TWPAs), one may still extract the visibility as would result from considering the full interferometer. We have emphasized this point a little more in the manuscript.

6 - While loss and in particular heating-induced losses are treated, I would expect the authors to mention other parasitic effects that might mislead one into thinking collapse had been observed. Are there other sources of phase noise to worry about? Have they been characterized elsewhere?

TvdR: We have not considered other sources of noise. Before interpreting results from an actual experiment these sources of noise would need to be characterised and taken into account in the calculation. This remark has been added to the feasibility section.

7 - I would also have liked some discussion of the regime this experiment could probe, compared with past work on mesoscopic entanglement, including the work by Gisin et al. on "micro-macro entanglement," work in superconducting qubits, and work on atomic ensembles (perhaps others as well).

TvdR: We did a calculation, in which I estimated that for 40dB gain 2e5 electrons would partake in the superposition. Together these have a mass of 2e-25 kg and yield 1e5 degrees of freedom (assuming the transmission lines are 1D and the electrons move along the transmission line and are paired in Cooper pairs). Current record for atomic ensembles: Gerlich et al. Nat. Commun. 2:263, 7e-24 kg with 1000 degrees of freedom. We then would need to add the superposition distance which we can estimate as a photon wavelength ~3mm in common microwave lines.
In the introduction we emphasized that our proposal is different from experiments exploiting optical photons, such as those by Zeilinger, Gisin, De Martini and Rempe. Optical photons in non-linear crystals will not lead to a superposition of mass, since the electrons in the crystal are effectively still localized (or rather de-localised over an atomic distance of ~0.1nm. In transmission lines the materials valence electrons that carry the photonic excitation are delocalized over the photon wavelength in the order of millimetres.

Report
I think this is a very interesting proposal, and the authors have made a good case. I would like to see the paper published, but think it is almost essential that they supplement the calculations with some physical discussion of the effects and the limits they treat.
I think is indispensible that they cite -- and actually discuss -- the earlier work (& controversy) on entanglement generated by parametric amplification.
Understanding the position-dependence of the coherent-state collapse model is, to me, central to the point of this paper. Perhaps there is no error in their result about the 0% visibility at the midpoint -- but if this is so, then the fact that I had the misapprehension I did still convinces me that the discussion as it stands should be bolstered by physical argument.
I think it is also essential that the paper be clear about what regimes have been probed in other experiments and how this regime differs (at least qualitatively); and that more discussion is provided of what culprits the authors might proceed to search for if they did carry out the experiment and find lower visibility than in the idealized quantum model.

Report 2
1. A highly speculative and unjustified conjecture for spontaneous collapse.
2. No attempt is made to determine if this conjecture is consistent with the results of decades of previous experiments in quantum optics

Report
The authors consider a simple optical device based on a single-photon Mach-Zehnder (MZ) interferometer but with identical parametric amplifiers placed in each internal path. They then calculate the visibility for the signal and idler interference visibility (for zero phase shift) at the outputs. This calculation is straightforward (despite the complicated method the authors use) in quantum optics and does not warrant publication. It is trivial to show that the visibility for the idler is constant at 1/3 while that for the signal cosh^2(r)/(cosh^2(r)+2 sinh^2(r)) as the authors show by the red lines in Fig 2. Inclusion of loss is also quite straightforward.

TvdR: Yes, the calculation is straightforward, but we are not aware that it has actually been performed. Moreover, since our quantum optics colleagues could not tell us the result straightaway, we doubt whether the result is trivial, also since it involves multiphoton interference and because temperature is more important for microwave paramps than for optical paramps.

The substantive part of the paper is based on their own model of 'spontaneous collapse'. This is speculative and rather vague. The only justification they offer is a claim that measurement requires amplification and so amplification must lead to wave-function collapse.

TvdR: We agree with the referee that we do not know whether wavefunction collapse will actually will occur in our proposed experiment. That is why we write “if collapse occurs …” the interference visibility of our proposed interferometer diminished.

Despite this rather dubious reasoning, there is no evidence for it despite the countless experiments that have been done on optical parametric amplifiers in quantum optics over decades. These devices are well modelled by unitary transformations as used by the authors.

TvdR: Optical parametric amplifiers and microwave paramps work in a totally different regime of the EM-spectrum. In OPAs the electrons are still bound to their atoms, and therefore there superposition distance is small (approximately the size of their orbital radius). On the other hand, in microwave PAs the valence electrons (Cooper pairs) move freely through the material and the passage of a single photon will superpose them over a distance in the order of the photon wavelength (approximately 3mm). To stress this point we added a remark to the introduction citing work of Gisin et al..
We believe that because microwave paramp experiments have become technologically feasible and because they work in a different regime than optical paramps, they are worth doing.

Loss is also easily included and well understood. If one puts a paramp inside a Fabry-Perot cavity (an interferometer) one gets a parametric oscillator; a common quantum optical device well described by conventional theory.
Before I address the author’s own rather unique model of 'spontaneous collapse', let me first object to the use of this term. To most of us the term spontaneous collapse refers to a detailed proposal from Ghirardi, Rimini and Weber. As far as I can see this bears no relation to the author’s definition of the term.

TvdR: GRW (and CSL etc.) use spontaneous collapse for a differential equation that describes how a quantum state might evolve in a classical state in time by introducing (mostly) two parameters: the collapse rate and the width of the state after collapse. It is generally assumed that collapse takes place towards the position basis. Although we do not use the differential of GRW equation explicitly, the result of the differential equation describes a collapsed state. Our spontaneous collapse is just a short-cut for this equation, in which case - as remarked below - the spontaneous collapse is described by a projection operator.

The model of spontaneous collapse used in this paper inserts two kinds of projection operators at various points in the internal arms of the MZ. One projects onto number states and the other projects onto coherent states. Obviously this will change the visibility. One could justify this description in terms of actual measurements taking place in each arm of the MZ: non-absorption counting in the case of photon number and heterodyne detection in the case of coherent states. The authors don’t mention this so presumably they mean something else by spontaneous collapse. (At some point they do refer to ‘which path’ measurements although there is no such measurement included in the model).

TvdR: we do justify the description, only not in these terms: collapse to a number state could take place due to a which path detection of the state, or due to some other mechanism. Collapse to a coherent state might take place, because the electrons in the transmission lines collapse into position states (classical motion can always be described by Fourier components, i.e. sinusoids, i.e. coherent states which are the closest quantum analogue to sinusoids).

I see little merit in publishing a proposal for an experiment based on such speculative and, given previous experiments, highly unlikely conjectures.
A minor point: The first sentence in the abstract is wrong. There are many experiments that project onto non classical states. See for example the Haroche experiments in the microwave regime.

TvdR: We changed the first sentence of the abstract from “… a
measurement causes a wave function to collapse into a well-defined classical state” to “… a measurement causes a wave function to collapse onto an eigenstate of the measurement apparatus”.

The first sentence of the paper is not wrong but only describes a very small class of actual experiments. Again, examples can be found in the experiments of Haroche.

TvdR: As far as we know, Haroche only did experiments on environmental decoherence, which is not what we are after.

---

## Round 4 · Author Response

Dear Editor-in-charge,
We hereby resubmit our paper “An experimental proposal to study collapse of the wave function in travelling-wave parametric amplifiers” in response to the referee reports submitted so far.

We have chosen to simplify the message of the paper. To avoid the readers of the paper to think in line of the spontaneous collapse theories available to date, we have chosen a more pragmatic line of reasoning independent from the available spontaneous collapse theories.

If one considers a microwave qubit experiment, we know that the qubit can be in a superposition of states, e.g. resulting in a superposition containing zero and one photon. If this state is measured, we always measure either zero or one photon, implying the state has collapsed at some point in between the qubit and the output of the measurement apparatus. In our proposed experiment we want to investigate at which point this happens.
Typically, the qubit is connected to the measurement apparatus by an amplification chain. As a first stage we therefore want to know whether the superposed state survives amplification, or that the collapse happens during the amplification process.

We hope that this simplification in the motivation for our paper will avoid the reader to run into unintended obstacles and that this satisfies you and the referees so that you can decide to publish our manuscript.

Sincerely,
Tom van der Reep
Louk Rademaker
Tjerk Oosterkamp

---

## Round 4 · List of Changes

We rewrote the introduction and abstract along the lines of the letter above. Throughout the paper we removed references to spontaneous collapse theories, as these do not contribute to our pragmatic line of reasoning and they distract from the point that we want to make. Furthermore, we added a comment on the stochasticity of the collapse as brought up by referee report 3. Without performing the actual calculation, we gave a sketch of what such a calculation and its expected result would look like at the end of section 6.

---

## Editorial Decision

editor-in-charge_assigned